# Diffusing DeBias:
# Synthetic Bias Amplification for Model Debiasing

**Massimiliano Ciranni**[1*]   **Vito Paolo Pastore**[1,2*]   **Roberto Di Via**[1*]
**Enzo Tartaglione**[3]   **Francesca Odone**[1]   **Vittorio Murino**[2,4]

[1]MaLGa-DIBRIS, University of Genoa, Italy
[2]AI for Good (AIGO), Istituto Italiano di Tecnologia, Genova, Italy
[3]Telècom-Paris, Ecole Polytechnique Superior, France
[4]Department of Computer Science, University of Verona, Italy

`{massimiliano.ciranni, roberto.divia}@edu.unige.it`, `vito.paolo.pastore@unige.it`
`enzo.tartaglione@telecom-paris.fr`, `francesca.odone@unige.it`, `vittorio.murino@iit.it`

## Abstract

The effectiveness of deep learning models in classification tasks is often challenged by the quality and quantity of training data whenever they are affected by strong spurious correlations between specific attributes and target labels. This results in a form of bias affecting training data, which typically leads to unrecoverable weak generalization in prediction. This paper addresses this problem by leveraging bias amplification with generated synthetic data only: we introduce Diffusing DeBias (DDB), a novel approach acting as a plug-in for common methods of unsupervised model debiasing, exploiting the inherent bias-learning tendency of diffusion models in data generation. Specifically, our approach adopts conditional diffusion models to generate synthetic bias-aligned images, which fully replace the original training set for learning an effective bias amplifier model to be subsequently incorporated into an end-to-end and a two-step unsupervised debiasing approach. By tackling the fundamental issue of bias-conflicting training samples' memorization in learning auxiliary models, typical of this type of technique, our proposed method outperforms the current state-of-the-art in multiple benchmark datasets, demonstrating its potential as a versatile and effective tool for tackling bias in deep learning models. Code is available at `https://github.com/Malga-Vision/DiffusingDeBias`.

## 1   Introduction

Deep learning models have shown impressive results in various vision tasks, including image classification. However, their success is highly dependent on the quality and representativeness of the training data. When a deep neural network is trained on a biased dataset, so-called "shortcuts", corresponding to spurious correlations (*i.e.*, *biases*) between irrelevant attributes and labels, are learned instead of semantic class attributes, affecting model generalization and impacting test performances. In other words, the model becomes biased towards specific training sub-populations presenting biases, known as *bias-aligned*, while samples not affected by the bias are referred to as *bias-conflicting* or unbiased. In this scenario, where it is generally assumed that most of the training samples available in a biased data set are biased (typically, $95\%$ or higher [39]), several debiasing methods have been proposed addressing the problem from different points of view [23, 46, 41, 22, 29]. When annotations on bias attributes are available, common *supervised* debiasing strategies involve reweighting the training set to give more importance to the few available bias-conflicting data [29], for example, by upsampling [31] or upweighting [43] them. Conversely, *unsupervised* debiasing methods, which

---

*These authors contributed equally to this work.

39th Conference on Neural Information Processing Systems (NeurIPS 2025).

assume no prior availability of bias information, typically involve an *auxiliary model* to estimate and convey information on bias for reweighting the training set, employing its signal to guide the training of a debiased target model, while diminishing its tendency to learn bias shortcuts.

To this end, there are several existing works [39, 23, 29, 44] exploiting *intentionally biased* auxiliary models, trained to overfit bias-aligned samples, thus being very confident in both correctly predicting bias-aligned training samples as well as misclassifying bias-conflicting ones [29]. Such a distinct behavior between the two subpopulations is required to convey effective bias supervisory signals for target model debiasing, including gradients or per-sample loss values, which, *e.g.*, should ideally be as high as possible for bias-conflicting and as low as possible for bias-aligned [39]. However, such desired behavior is severely hindered by bias-conflicting training samples, despite their low number. Indeed, the low cardinality of this population impedes a suitable modeling of the correct data distribution, yet, they are sufficient to harm the estimation of a reliable bias-aligned data distribution, despite the latter having much higher cardinality. This scenario occurs under two main conditions. First, bias-conflicting samples in the training set act as noisy labels, affecting the bias learning for the auxiliary models [29]. Second, these models tend to overfit and memorize bias-conflicting training samples within very few epochs, failing to provide different learning signals for the two subpopulations [9]. Despite the available strategies, including the usage of the Generalized Cross Entropy (GCE) loss function [39, 41], ensembles of auxiliary models [23, 29], or even *ad hoc* solutions tailored to the specific dataset considered (*e.g.*, training for only one epoch or assuming the availability of a bias-annotated validation [35]), how to define a protocol for effective and robust auxiliary model training for unsupervised debiasing remains an open issue.

Remarkably, recent works [41, 29, 23] underline how the target model generalization is profoundly impacted by the protocol used to train such an auxiliary model. In principle, if one could remove all bias-conflicting samples from the training set, it would be possible to train a bias-capturing model robust to bias-conflicting samples' memorization and interference, and capable of conveying ideal learning signals [29].

To this aim, our intuition is to elude these drawbacks by skipping the use of the actual training data in favor of another set of samples synthetically generated by an adequately trained diffusion model. Specifically, we propose to circumvent bias-conflicting interference on auxiliary models leveraging *synthetic biased* data obtained by exploiting Conditional Diffusion Probabilistic Models (CDPMs). We aim to learn a *per-class biased* image distribution, from which we sample an amplified synthetic bias-aligned subpopulation. The resulting synthetic bias-aligned samples are then exploited for training a *Bias Amplifier* model, instead of using the original actual training set, solving the issue of bias-conflicting overfitting and interference by construction, as the bias-capturing model does not see the original training set *at all*. We refer to the resulting approach as *Diffusing DeBias* (DDB). The intuition is indirectly supported by recent works studying the problem of biased datasets for image generation tasks [8, 24], and showing how diffusion models' generations can be biased, as a consequence of bias present in the training sets (See Sec. 2).

This tendency can be seen as an undesirable behavior of generative models, which affects the fairness of generated images, and recent articles have started to propose solutions to decrease the inclination of diffusion models to learn biases [21, 24, 42]. However, we claim that what is considered a drawback for generative tasks can be turned into an advantage for image classification's unsupervised model debiasing, as it allows the training of a robust auxiliary model that can ideally be plugged into various debiasing schemes, solving bias-conflicting training samples memorization by construction. For this reason, we refer to DDB as a plug-in for debiasing methods in image classification. To prove its effectiveness and versatility, we incorporate DDB's bias amplifier into two different debiasing frameworks, which we refer to as *Recipe I* and *Recipe II*, including a two-step and an end-to-end approach (see Sec. 2). Specifically, we exploit our bias-amplifier to: (i) extract subpopulations further used as pseudo-labels within the popular G-DRO [43] algorithm (Recipe I, Sec. 3.3.1), and (ii) provide a loss function for the training set to be used within an end-to-end method (Recipe II, Sec. 3.3.2). Both approaches are competitive with the state-of-the-art on popular benchmark datasets. In summary, our main contributions can be regarded as follows:

- We introduce DDB, a novel and effective debiasing framework that exploits CDPMs to learn *per-class* bias-aligned distributions in biased datasets with unknown bias information. The resulting CDPM is used to generate synthetic bias-aligned images later used to train a robust bias amplifier model, thus avoiding real bias-conflicting training sample memorization and construction interference (Sec. 3).

- We propose the usage of DDB as a versatile plug-in for debiasing approaches, designing two *recipes*: one based on a two-step procedure (Sec. 3.3.1), and the other as an end-to-end (Sec. 3.3.2) algorithm. Both proposed approaches prove to be effective, achieving state-of-the-art results (Sec. 4) across popular benchmark datasets, characterized by different types of single or multiple biases.

## 2   Related Work

In this section, we provide a broad categorization of model debiasing works.

**Supervised Debiasing.** Supervised methods exploit available prior knowledge about the bias (*e.g.*, bias attribute annotations), indicating whether a sample exhibits a particular bias (or not). A common strategy consists of training a *bias classifier* to predict such attributes so that it can guide the *target* model to extract unbiased representations [1, 48]. Alternatively, bias labels can be thought of as pre-defined encoded subgroups among target classes, allowing for the exploitation of robust training techniques such as G-DRO [43]. At the same time, bias information can be employed to apply regularization schemes aiming at forcing invariance over bias features [3, 45].

**Unsupervised Debiasing.** It considers bias information unavailable (the most likely scenario in real-world applications), thus having the highest potential impact among the described paradigms. Among such methods, a popular strategy consists in involving auxiliary models at any level of the debiasing process. We categorize these methods into two-step [35, 44, 22], and end-to-end [39, 33], in the following paragraphs.

**End-to-end approaches.** Here, bias mitigation is performed online with dedicated optimization objectives. Employing techniques such as ensembling and noise-robust losses has been proposed, though they require careful tuning of all the hyperparameters, which may require the availability of validation sets with bias annotations to work. In Learning from Failure (LfF), Nam *et al.* [39] assume that bias features are learned earlier than task-related ones: they employ a *bias-capturing* model to focus on easier (bias-aligned) samples through the GCE loss [53]. At the same time, a debiased network is jointly trained to assign larger weights to samples that the bias-capturing model struggles to discriminate. In DebiAN, Li *et al.* [33] jointly train an auxiliary model (the *discoverer*) and a *classifier* (the target model) in an alternate scheme so that the discoverer can identify bias learned by the classifier during training, optimizing an ad-hoc loss function for bias mitigation.

**Two-step methods.** Here, the auxiliary model is employed in a first step to identify bias in training data, generally through pseudo-labeling, while the second stage consists of actual bias mitigation exploiting the inferred pseudo-labels. In [35], the auxiliary model is trained with standard ERM and then used in inference on the training set, considering misclassified samples as bias-conflicting and vice-versa: debiasing is brought on by up-sampling the predicted bias-conflicting samples. Notably, they must rely on a validation set with bias annotations to stop the auxiliary model's training after a few epochs to avoid memorization and to tune the most effective upsampling factor. In [23], a set of auxiliary classifiers is ensembled (*bootstrap ensembling*) into a *bias-commmittee*, proposing that debiasing can be performed using a weighted ERM, where the weights are proportional to the number of models in the ensemble misclassifying a certain sample. Finally, a recent trend involves the usage of vision-language models to identify bias, further exploiting bias predictions to mitigate biases in classification models [10, 25, 52].

Among the described methods, some exploit the auxiliary model for bias amplification ([29, 39, 44]) under the general assumption that a bias amplification model would either misclassify or be less confident on bias-conflicting samples. In this case, the final debiasing performance is heavily dependent on such behavior, and specifically, on *how strongly* the auxiliary model captures the bias in training data [41, 29]. However, this fundamental assumption only holds if bias-conflicting samples are not memorized during training, which is likely to happen in very few training epochs [9, 41], especially considering that the very few training bias-conflicting samples are more likely to be overfitted, as reported in [43, 5]. One could argue that an annotated validation set could be used for properly regularizing the bias amplification model, but the latter cannot be assumed available a priori in real cases, being this an emerging argument of discussion among the community [9, 41]. For this reason, differently from the previously cited research, we propose in this work that synthetic bias-aligned samples could be used *instead of* the original training set, solving the bias-conflicting

memorization issue by construction. However, a real-world scenario using data with unknown bias poses a problem for generating a *pure* distribution of bias-aligned samples. As a solution, we propose to leverage diffusion models for generating a synthetic bias-aligned training set to be used for training a bias amplifier.

While the vast majority of available works regarding generative models and debiasing refer to strategies for obtaining unbiased synthetic data generation [4, 11], to the best of our knowledge, we are the first to propose the use of generative models for obtaining an amplified biased distribution. The closest work employing generative models for model debiasing is Jung *et al.* [19], where a debiasing method exploiting a GAN [12] is introduced, which is trained for style-transfer between several *biased* appearances with different objectives. Here, a target model is debiased through contrastive learning between images affected by different biases for bias-invariant representations. Similar strategies, where bias-conflicting samples are augmented to support model debiasing, can be found in a few other works [34, 18, 28]. In [28], the problem of bias-conflicting memorization is stressed, proposing a bias-unsupervised strategy to augment and increase the diversity of bias-conflicting features, exploiting a bias amplifier analogously to [39] for their identification, while improving the generalization of a target debiased model. In [34], an intentionally biased auxiliary model is exploited, which is trained on the biased dataset, supplementing bias-conflicting samples for the target debiased model. Here, images are generated by solving an optimization problem consisting of attacking the auxiliary model predictions while preserving the ones of the target debiased model. In [18], bias-conflicting samples are again augmented exploiting the popular mix-up approach [50]. Specifically, an auxiliary model is employed to amplify easy-to-learn features, and mix-up pairs with either the same ground truth label, but different biased features (evaluated with clustering in the feature space of an intentionally biased model), or samples with different biased features (*i.e.*, belonging to different clusters), but showing the same ground truth label. Other works have also explored generative models for debiasing. For instance, BiaSwap [22] utilizes image translation techniques to swap bias attributes between images, aiming to create more balanced training data. However, methods that attempt to create synthetic bias-conflicting samples [22, 34] may require high semantic fidelity, while our method creates a functionally biased dataset, which we believe to be a simpler and more robust objective.

Differently from the previously described approaches, we are proposing an alternative strategy to avoid bias-conflicting memorization, while significantly improving the generalization of the target debiased model. In our work, we show how it is possible to leverage the intrinsic tendency of diffusion models to capture bias in data, using conditioned generations to infer the biased distribution of each target class, thus allowing the sampling of new synthetic images, sharing the same bias pattern of the original training data. Our method designs a memorization-free bias amplified model, capable of providing precise supervisory signals for accurate model debiasing, virtually compatible with any debiasing approach involving auxiliary models.

## 3 The Approach

Our proposed debiasing approach is schematically depicted in Figure 1. In this Section, we first describe the general problem setting (Sec. 3.1), then detail DDB's two main components: *bias diffusion* (Sec. 3.2) and the two *Recipes* for model debiasing (Sec. 3.3).

### 3.1 Problem Setting

Let us consider a general data distribution $p_{\text{data}}$, typically encompassing multiple factors of variation and classes, and to build a dataset of images with the associated labels $\mathcal{D} = \{(\mathbf{x}_i, y_i)\}_{i=1}^N$ sampled from such a distribution. Let us also assume that the sampling process to obtain $\mathcal{D}$ is not uniform across latent factors of variations, *i.e.*, possible biases such as context, appearance, acquisition noise, viewpoint, etc.. In this case, data will not faithfully capture the true data distribution ($p_{\text{data}}$) just because of these bias factors. This phenomenon deeply affects the generalization capabilities of deep neural networks in classifying unseen examples not present the same biases. In the same way, we could consider $\mathcal{D}$ as the union between two sets, *i.e.* $\mathcal{D} = \mathcal{D}_{\text{unbiased}} \bigcup \mathcal{D}_{\text{biased}}$. Here, the elements of $\mathcal{D}_{\text{unbiased}}$ are uniformly sampled from $p_{\text{data}}$ and, in $\mathcal{D}_{\text{biased}}$, they are instead sampled from a conditional distribution $p_{\text{data}}(\mathbf{x}, y \mid b)$, with $b \in B$ being some latent factor (bias attribute) from a set of possible attributes $B$, likely to be unknown or merely not annotated, in a realistic setting [24]. If $|\mathcal{D}_{\text{biased}}| \gg |\mathcal{D}_{\text{unbiased}}|$, optimizing a classification model $f_\theta$ over $\mathcal{D}$ likely results in biased predictions

and poor generalization. This is due to the strong correlation between $b$ and $y$, often called *spurious correlation*, and denoted as $\rho(y, b)$, or just $\rho$ for brevity [22, 43, 38]), which is dominating over the true target distribution semantics.

It is important to notice that data bias is a general problem, not only affecting classification tasks but also impacting several others such as data generation [8]. For instance, given a Conditional Diffusion Probabilistic Models (CDPM) modeled as a neural network $\tilde{g}_\phi(\mathbf{x} \mid y)$ (with parameters $\phi$) that learns to approximate a conditional distribution $p(\mathbf{x} \mid y)$ from $\mathcal{D}$, we expect that its generations will be biased, as also stated in [8, 24]. While this is a strong downside for image-generation purposes, in this work, we claim that when $\rho(y, b)$ is very high (*e.g.* $\geq 0.95$, as generally assumed in model debiasing literature [39]), a CDPM predominantly learns the biased distribution of a specific class, *i.e.*, $\tilde{g}_\phi(\mathbf{x} \mid y) \approx p(\mathbf{x} \mid b)$ rather than $p(\mathbf{x} \mid y)$.

## 3.2 Diffusing the Bias

In the context of mitigating bias in classification models, the tendency of a CDPM to approximate the per-class biased distribution represents a key feature for training an auxiliary *bias amplified* model.

**The Diffusion Process.** The diffusion process progressively converts data into noise through a fixed Markov chain of $T$ steps [15]. Given a data point $\mathbf{x}_0$, the forward process adds Gaussian noise according to a variance schedule $\{\beta_t\}_{t=1}^T$, resulting in noisy samples $\mathbf{x}_1, \ldots, \mathbf{x}_T$. This forward process can be formulated for any timestep $t$ as: $q(\mathbf{x}_t|\mathbf{x}_0) = \mathcal{N}(\mathbf{x}_t; \sqrt{\bar{\alpha}_t}\mathbf{x}_0, (1 - \bar{\alpha}_t)\mathbf{I})$, where $\bar{\alpha}_t = \prod_{s=1}^t \alpha_s$ with $\alpha_s = 1 - \beta_s$. The reverse process then gradually denoises a sample, reparameterizing each step to predict the noise $\epsilon$ using a model $\epsilon_\theta$:

$$\mathbf{x}_{t-1} = \frac{1}{\sqrt{\alpha_t}} \left( \mathbf{x}_t - \frac{\beta_t}{\sqrt{1 - \bar{\alpha}_t}} \boldsymbol{\epsilon}_\theta(\mathbf{x}_t, t) \right) + \sigma_t \mathbf{z}, \tag{1}$$

where $\mathbf{z} \sim \mathcal{N}(\mathbf{0}, \mathbf{I})$ and $\sigma_t = \sqrt{\beta_t}$.

**Classifier-Free Guidance for Biased Image Generation.** In cases where additional context or *conditioning* is available, such as a class label $y$, diffusion models can use this information to guide the reverse process, generating samples that better reflect the target attributes and semantics. Classifier-Free Guidance (CFG) [16] introduces a flexible conditioning approach, allowing the model to balance conditional and unconditional outputs without dedicated classifiers.
The CFG technique randomly omits conditioning during training (*e.g.*, with probability $p_{\text{uncond}} = 0.1$), enabling the model to learn both generation modalities. During the sampling process, a guidance scale $w$ modulates the influence of conditioning. When $w = 0$, the model relies solely on the conditional model. As $w$ increases ($w \geq 1$), the conditioning effect is intensified, potentially resulting in more distinct features linked to $y$, thereby increasing fidelity to the class while possibly reducing diversity, whereas lower values help to preserve diversity by decreasing the influence of conditioning. The guided noise prediction is given by:

$$\boldsymbol{\epsilon}_t = (1 + w)\boldsymbol{\epsilon}_\theta(\mathbf{x}_t, t, y) - w\boldsymbol{\epsilon}_\theta(\mathbf{x}_t, t), \tag{2}$$

where $\boldsymbol{\epsilon}_\theta(\mathbf{x}_t, t, y)$ is the noise prediction conditioned on class label $y$, and $\boldsymbol{\epsilon}_\theta(\mathbf{x}_t, t)$ is the unconditional noise prediction. This modified noise prediction replaces the standard $\boldsymbol{\epsilon}_\theta(\mathbf{x}_t, t)$ term in the reverse process formula (Equation 1). In this work, we empirically show how CDPM learns and amplifies the underlying biased distribution when trained on a biased dataset with strong spurious correlations, allowing bias-aligned image generation.

## 3.3 DDB: Bias Amplifier and Model Debiasing

As stated in Sec. 2, a typical unsupervised approach to model debiasing relies on an auxiliary intentionally-biased model, named here as *Bias Amplifier* (BA). This model can be exploited in either 2-step or end-to-end approaches, denoted here as *Recipe I* and *Recipe II*, respectively. This section describes the two debiasing recipes we employ in this work to demonstrate the effectiveness of our BA as a plug-in for unsupervised model debiasing.

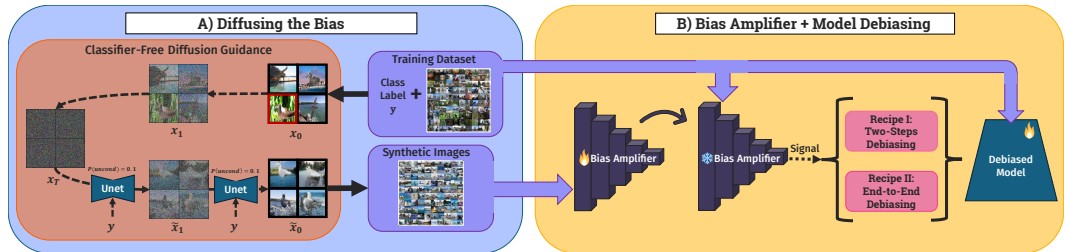

Figure 1: **Schematic representation of our DDB framework.** The debiasing process consists of two key steps: (A) *Diffusing the Bias* uses a conditional diffusion model with classifier-free guidance to generate synthetic images that preserve training dataset biases, and (B) employs a *Bias Amplifier* firstly trained on such synthetic data, and subsequently used during inference to extract supervisory bias signals from real images. These signals are used to guide the training process of a target debiased model by designing two *debiasing recipes* (*i.e.*, 2-step and end-to-end methods).

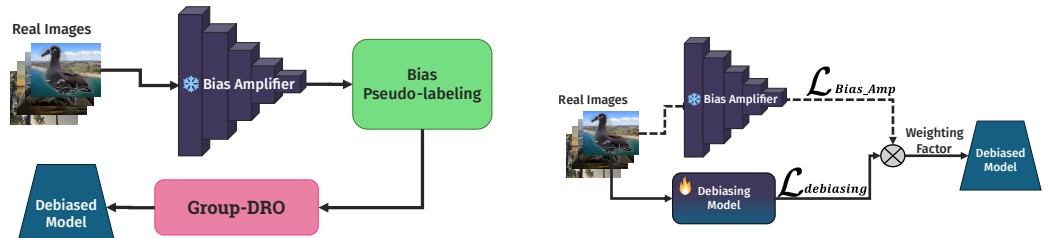

(a) Overview of *Recipe I*'s 2-step debiasing approach.

(b) Overview of *Recipe II*'s end-to-end debiasing approach.

Figure 2: Comparison of two debiasing strategies: *Recipe I* and *Recipe II*.

### 3.3.1 Recipe I: 2-step debiasing

The adopted 2-step approach consists of 1) applying the auxiliary model trained on biased generated data to perform a bias pseudo-labeling, hence estimating bias-aligned/bias-conflict split of original actual data, and 2) applying a *bias supervised* method to train a debiased target model for classification. For the latter, we use the group DRO algorithm [43] (G-DRO) as a proven technique for the pure debiasing step. In other words, being in the unsupervised bias scenario where the real bias labels are unknown, we estimate bias pseudo-labels performing an inference step by feeding the trained BA with the original actual training data, and identifying as bias-aligned the correctly classified samples, and as bias-conflicting those misclassified. Among possible strategies to assign bias pseudo-labels, such as feature-clustering [44] or anomaly detection [41], we adopt a simple heuristic based on the BA misclassifications. Specifically, given a sample $(\mathbf{x}_i, y_i, c_i)$ with $c_i$ unknown pseudo-label indicating whether $\mathbf{x}_i$ is bias conflicting or aligned, we estimate bias-conflicting samples as

$$\hat{c}_i = \mathbb{1}\left(\hat{y}_i \neq y_i \ \wedge \ \mathcal{L}(\hat{y}_i, y_i) \ > \ \mu_n(\mathcal{L}) + \gamma\sigma_n(\mathcal{L})\right), \tag{3}$$

where $\mathbb{1}$ is the indicator function, $\mathcal{L}$ is the CE loss of the BA on the real sample, and $\mu_n$ and $\sigma_n$ represent the average training loss and its standard deviation, respectively, depending on the loss $\mathcal{L}$. Together with the multiplier $\gamma \in \mathbb{N}$, this condition defines a sort of filter over misclassified samples, considering them as conflicting only if their loss is also higher than the mean loss increased by a quantity corresponding to a certain z-score of the per-sample training loss distribution ($\mu_n(\mathcal{L}) + \gamma\sigma_n(\mathcal{L})$ in Eq. 3). Once bias pseudo-labels over original training data are obtained, we plug in our estimate as group information for the G-DRO optimization, as schematically depicted in Figure 2a.

The above *filtering* operation refines the plain *error set*, restricting bias-conflicting sample selection to the hardest training samples, with potential benefits for the most difficult correlation settings ($\rho > 0.99$). Later in the experimental section, we provide an ablation study comparing different filtering ($\gamma$) configurations and plain error set alternatives.

### 3.3.2 Recipe II: end-to-end debiasing

A typical end-to-end debiasing setting includes the joint training of the target debiasing model and one [39] or more [23, 29] auxiliary intentionally-biased models. Here, we design an end-to-end

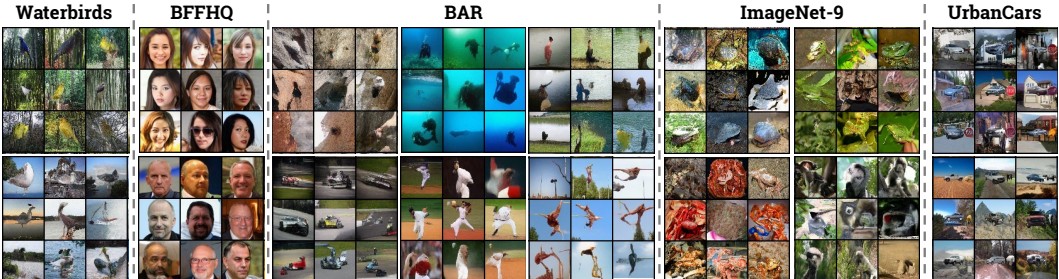

| Waterbirds | BFFHQ | BAR | ImageNet-9 | UrbanCars |

Figure 3: **Examples of synthetic bias-aligned images across multiple datasets.** Each grid shows synthetic images for specific classes across biased datasets, revealing how the model aligns with dataset-specific biases in different contexts.

debiasing procedure, denoted as *Recipe II*, incorporating our BA by customizing a widespread general scheme, introduced in the Learning from Failure (LfF) method [39]. LfF leverages an intentionally-biased model trained using Generalized CE (GCE) loss to support the simultaneous training of a debiased model adopting the CE loss re-weighted by a per-sample relative difficulty score. Specifically, we replace the GCE biased model with our Bias Amplifier, which is frozen and only employed in inference to compute its loss function for each original training sample ($\mathcal{L}_{\text{bias\_amp}}$), as schematically represented in Figure 2b. Such a loss function is used to obtain a weighting factor for the target model loss function, defined as $r = \frac{\mathcal{L}_{\text{Bias\_Amp}}}{\mathcal{L}_{\text{debiasing}} + \mathcal{L}_{\text{Bias\_Amp}}}$. Coarsely speaking, $r$ should be low for bias-aligned and high for bias-conflicting samples.

## 4 Experiments

In this Section, we report the description of the benchmark datasets employed, followed by a discussion of the obtained results and a thorough ablation analysis. In the Appendix, we report the general implementation (Sec. A.2) and training details, including the implementation details of *Recipe I* (Sec. A.3) and *Recipe II* (Sec. A.4), more examples and analysis on the generated images (Sec. A.6), and additional debiasing results on less biased settings for Waterbirds plus an alternative version of BAR having known $\rho$ and bias-annotations.

### 4.1 Datasets

Our experiments utilize six distinct datasets: three typical benchmark datasets, including Waterbirds [43], Biased Flickr-Faces-HQ (BFFHQ) [22], and Biased Action Recognition (BAR), taken from the original version as introduced in [39]. On top of these benchmark datasets, we are interested in evaluating our approach in challenging scenarios involving real-world images with natural biases and datasets where multiple shortcuts are present. For this reason, we include ImageNet-9 [2]/ImageNet-A [14], and UrbanCars [32] in our experiments. More dataset details are in Sec. A.1.

### 4.2 Results

#### 4.2.1 Synthetic bias-aligned image generation

First, we present qualitative image generation results across multiple datasets (see Fig. 3) to validate the effectiveness of our bias diffusion approach. The generated samples maintain good fidelity and capture the typical bias patterns of each training set per class, confirming the conditioned diffusion process's ability to learn and diffuse inherent biases. This is confirmed by screening the generated images by three independent annotators, who were asked to verify (or deny) the presence of the bias in each synthetic sample, where possible. Specifically, for Waterbirds, the reported average bias-aligned proportion of synthetic generated images is 98.35%, in BAR it is 99.20%, and 99.70% is obtained for BFFHQ. All these values are higher than the known proportions of bias in the original datasets. In addition, focusing on Waterbirds, we also evaluated the generated image content by adopting an image captioner, specifically BLIP-2 [30]. We estimate a caption for each image per

class and count the frequency of every word. Supplementary Material (Sec. A.8) shows a histogram of the detected keywords, in which it can be noted that together with a high frequency of the object class (*bird*), similar significant bins of the bias attributes (*land*, *water*) are also evidenced.

#### 4.2.2 Classification accuracy

Tables 1a, 1b, 2a, 2b, and 3 report the average and standard deviation of our employed metrics, obtained over three independent runs for all the considered benchmark datasets. We dedicate a separate table for each dataset, as not all methods share the same benchmarks. We report Worst Group Accuracy (WGA) for Waterbirds, as in [43, 29]. For BFFHQ, we report the average and the conflicting accuracy (*i.e.*, the average accuracy computed only on the bias-conflicting test samples), following [22, 33]. BAR and ImageNet-A do not provide bias-attribute annotations, hence, we report just the average test accuracy. For simplicity, throughout this section, we will refer to *Recipe I* and *Recipe II* as DDB-I and DDB-II, respectively. It is worth noticing that, for BAR and BFFHQ, we report benchmark results related to works that employ the same versions as ours. Across all the considered

Table 1: Results of DDB Recipes I and II on Waterbirds (left) and BAR (right). Ticks (✓) mark bias-unsupervised methods. Best unsupervised results are in bold; second-best are underlined.

(a) Debiasing results on Waterbirds

| Method | Unsup | Waterbirds WGA |
|---|---|---|
| ERM | – | $62.60 \pm 0.30$ |
| LISA [49] | – | $89.20$ |
| G-DRO [43] | – | $91.40 \pm 1.10$ |
| George [44] | ✓ | $76.20 \pm 2.00$ |
| JTT [35] | ✓ | $83.80 \pm 1.20$ |
| CNC [51] | ✓ | $88.50 \pm 0.30$ |
| B2T+G-DRO [25] | ✓ | $90.70 \pm 0.30$ |
| LfF [39] | ✓ | $78.00$ |
| CDvG+LfF [19] | ✓ | $84.80$ |
| MoDAD [41] | ✓ | $89.43 \pm 1.69$ |
| DDB-II (ours) | ✓ | $\mathbf{91.56} \pm 0.15$ |
| DDB-I (ours) | ✓ | $\underline{90.81} \pm 0.68$ |
| DDB-I (w/ err. set) | ✓ | $90.34 \pm 0.41$ |

(b) Debiasing results on BAR

| Method | Unsup | BAR Avg. |
|---|---|---|
| ERM | – | $51.85 \pm 5.92$ |
| BiaSwap [22] | ✓ | $52.40 \pm$ - |
| JTT [35] | ✓ | $68.53 \pm 3.29$ |
| LfF [39] | ✓ | $62.98 \pm 2.76$ |
| LWBC [23] | ✓ | $62.03 \pm 2.76$ |
| DebiAN [33] | ✓ | $69.88 \pm 2.92$ |
| MoDAD [41] | ✓ | $69.83 \pm 0.72$ |
| DDB-II (ours) | ✓ | $\mathbf{72.81} \pm 1.02$ |
| DDB-I (ours) | ✓ | $70.40 \pm 1.41$ |
| DDB-I (w/ err. set) | ✓ | $\underline{70.59} \pm 0.19$ |

benchmark datasets, both DDB recipes show state-of-the-art results. Notably, in Waterbirds, DDB's WGA is higher than all the other state-of-the-art unsupervised methods' accuracies, as well as of that of the supervised G-DRO [43] (91.56% vs. 91.40% of G-DRO). DDB-I is providing slightly worse performance than DDB-II, but still better than the state of the art. In BAR, DDB-II is reaching

Table 2: Results of DDB Recipes I and II on BFFHQ (left) and ImageNet-9/A (right) from [22]. Ticks (✓) mark bias-unsupervised methods. Best unsupervised results are in bold; second-best are underlined.

(a) Debiasing results on BFFHQ

| Method | Unsup | BFFHQ Avg. | Confl. |
|---|---|---|---|
| ERM | – | – | $60.13 \pm 0.46$ |
| BiaSwap [22] | ✓ | – | $58.87 \pm$ - |
| JTT [35] | ✓ | – | $62.20 \pm 1.34$ |
| LfF [39] | ✓ | – | $62.97 \pm 3.22$ |
| ETF-Debias [47] | ✓ | – | $73.60 \pm 1.22$ |
| Park et al. [40] | ✓ | $71.68$ | – |
| CDvG+LfF [19] | ✓ | – | $62.20 \pm 0.45$ |
| DebiAN [33] | ✓ | – | $62.80 \pm 0.60$ |
| MoDAD [41] | ✓ | – | $68.33 \pm 2.89$ |
| DDB-II (ours) | ✓ | $\mathbf{83.15} \pm 1.76$ | $70.93 \pm 0.14$ |
| DDB-I (ours) | ✓ | $81.27 \pm 0.88$ | $\mathbf{74.67} \pm 2.37$ |
| DDB-I (w/ err. set) | ✓ | $\underline{82.44} \pm 0.64$ | $71.40 \pm 0.92$ |

(b) Debiasing results on ImageNet-9/A

| Method | Unsup | ImageNet-A Avg. |
|---|---|---|
| ERM | – | $30.30$ |
| LWBC [23] | ✓ | $35.97 \pm 0.49$ |
| CDvG+LfF [19] | ✓ | $34.60$ |
| DDB-II (ours) | ✓ | $37.53 \pm 0.82$ |
| DDB-I (ours) | ✓ | $\mathbf{39.80} \pm 0.50$ |
| DDB-I (w/ err. set) | ✓ | $\underline{38.12} \pm 0.96$ |

72.81% average accuracy, which is 2.93% better than the second best (DebiAN [33]), in terms of conflicting accuracy. In BFFHQ, our DDB-I reaches an accuracy of 74.67%, with an average improvement of 1% when compared to the state-of-the-art ETF-Debias [47] (73.60%). For UrbanCars

Table 3: Comparison of deep learning methods on the UrbanCars biased dataset. ID.ACC is in-distribution accuracy (the higher, the better); other three metrics show the accuracy gap between each minority group and ID.ACC (the lower, the better). GroupDRO and LLE are separated as they rely on bias information.

| Method | ID.ACC ↑ | BG-Gap ↓ | CoObj-Gap ↓ | BG-CoObj-Gap ↓ |
|---|---|---|---|---|
| GroupDRO [43] | 91.60 | 10.90 | 3.60 | 16.40 |
| LLE [32] | 96.70 | 2.10 | 2.70 | 5.90 |
| LfF [39] | 97.20 | 11.60 | 18.40 | 63.20 |
| JTT [35] | 95.90 | 8.10 | 13.30 | 40.10 |
| EIIL [7] | 95.50 | 4.20 | 24.70 | 44.90 |
| DebiAN [33] | 98.00 | 14.90 | 10.50 | 69.00 |
| DDB-I (ours) | 86.39±0.74 | **1.85**±3.21 | **0.52**±1.38 | **0.12**±1.56 |
| DDB-II (ours) | **98.56**±0.92 | 2.30±0.60 | 11.10±1.20 | 46.70±2.42 |

(see Table 3), presenting multiple biases, we adopted the same metrics of [32] (see Suppl. Material, Sec. A.1). DDB-I shows overall the most balanced performance across bias-conflicting samples, reporting an improvement over LLE [32] of $0.25\%$ for BG-Gap, $2.18\%$ on CoObj-Gap, and $5.78\%$ for BG-CoObj-Gap. DDB-II provides a significant improvement over the original LfF, but struggles more with the background-conflicting samples than DDB-I.

## 4.3 Ablation analysis

In this section, we provide an extensive ablation analysis of DDB main components. Without losing generality, all ablations are performed on Waterbirds and exploit Recipe I for model debiasing. Additional ablations on Recipe II can be found in the Supp. Mat. The reported analyses regard several aspects of the data generation process with respect to the Bias Amplifier, the effects of the filtering option in Recipe I, and the behavior of DDB on unbiased data.

**Bias Amplifier data requirements.** All our main results are obtained by training a BA on 1000 synthetic images per class, and we want here to measure how this choice impacts its effectiveness. From Table 4 (left), we observe that even 100 synthetic images are enough for good debiasing performance despite not being competitive with top accuracy scores. Increasing the number of training images improves WGA, reaching the best performance for 1000 images per class.

**CFG strength and generation bias.** The most important free parameter for the CDPM with CFG is the guidance conditioning *strength* $w$. Such strength can affect the variety and consistency of generated images. Here, we investigate DDB dependency on this parameter. Table 4 (right) shows a limited impact on debiasing performance. WGA variations along different $w$ values equal or larger than 1 are limited, with the best performance obtained for $w = 1$.

Table 4: Ablations on: (left) synthetic image count for training the BA on Waterbirds; (right) Impact of guidance strength $w$ on CDPM sampling, evaluated on Waterbirds. For both experiments, we employ *Recipe I*.

| # Synth. images for BA training | | | | | |
|---|---|---|---|---|---|
| **# Imgs.** | 100 | 200 | 500 | 1000 | 2000 |
| **WGA** | 86.25 | 87.10 | 87.67 | 89.05 | **90.81** |

| Guidance strength | | | | | |
|---|---|---|---|---|---|
| $w$ | 0 | **1** | 2 | 3 | 5 |
| **WGA** | 87.85 | **90.81** | 89.25 | 89.56 | 88.32 |

**Group extraction via BA plain error sets.** In Sec. 3.3.1, we describe how we filter the BA *error set* through a simple heuristic based on per-sample loss distribution. In Tables 1a, 1b, 2a, and 2b, the last row (DDB-I w/ plain err. set) reports our obtained results when considering as bias-conflicting all the samples misclassified by the Bias Amplifier. Our results show a trade-off between the two alternatives. The entire set of misclassifications is always enough to reach high bias mitigation performance with less critical settings (e.g., like in Waterbirds, $\rho = 0.950$), gaining an advantage from such a coarser selection. On the other hand, the importance of a more precise selection is highlighted by the superior performance of the filtered alternative in the more challenging scenarios, like in BFFHQ ($\rho = 0.995$).

**Bias-Identification Accuracy.** To frame the goodness of our BA in correctly identifying aligned and conflicting training samples, we provide the accuracy of such a process when considering samples misclassified by it. Specifically, we reach a high bias-identification accuracy of $89.00\%$ for Waterbirds and $96.00\%$ on BFFHQ, supporting the effectiveness of the proposed protocol, which is not dependent on any careful regularization, bias-annotated validation sets, or large ensembles of auxiliary classifiers.

**DDB impact on unbiased dataset.** In real-world settings, it is generally unknown whether bias is present in datasets or not. Consequently, an effective approach, while mitigating bias dependency in the case of a biased dataset, should also not degrade the performance when the bias is not present. To evaluate DDB in this regard, we apply the entire pipeline (adopting Recipe I) on a common image dataset such as CIFAR-10 [27], comparing it with traditional ERM training with CE loss. In this controlled case, *Recipe I* provides a test accuracy of $84.16\%$, with a slight improvement of $\sim 1\%$ compared to traditional ERM ($83.26\%$) employing the same target model (ResNet18). This result shows that our Bias Amplifier can still provide beneficial signals to the target model in standard scenarios, potentially favoring the learning of the most complex samples.

**Effectiveness of Synthetic Data for BA Training.** A core claim of our work is that training the Bias Amplifier on synthetic data prevents the memorization of real bias-conflicting samples. To validate this, we compare our standard BA (trained on synthetic data) with a BA trained on the original real training data within our two recipes. Using the real dataset significantly degrades performance. For Recipe I, the BA trained on real data for 50 epochs memorized all conflicting samples, leading to empty pseudo-label groups. Training for just one epoch (as in JTT [35]) yielded a WGA of only $79.43\%$. For Recipe II, using a frozen BA trained on real data resulted in a WGA of $78.45\%$, far below our proposed method and in line with the original LfF performance. These results empirically confirm that using synthetic data is crucial for creating an effective BA.

## 5 Conclusions

In this work, we present Diffusing DeBias (DDB), a novel debiasing framework capable of learning the per-class bias-aligned training data distribution, leveraging a CDPM. Synthetic images sampled from the inferred biased distribution are used to train a Bias Amplifier, employed to provide a supervisory signal for training a debiased model. The usage of synthetic bias-aligned data is empirically shown to prevent the detrimental effects typically impacting auxiliary models' training in unsupervised debiasing schemes, such as bias-conflicting sample overfitting and interference, which lead to suboptimal generalization performance of the debiased target model. Our experiments confirm that a bias amplifier trained on real data fails at this task, while our approach creates a robust and effective supervisory signal. DDB is versatile, acting as a plug-in for unsupervised debiasing methods, since its Bias Amplifier can be incorporated into several debiasing approaches. In this work, we incorporate DDB's Bias Amplifier in a two-step and an end-to-end debiasing recipes, outperforming state-of-the-art works on popular biased benchmark datasets (considering single and multiple biases), including recent works relying on vision-language models [25, 6]. Finally, while effectively mitigating bias dependency for biased datasets, DDB does not degrade performance when used with unbiased data, making it suitable to be adopted as an effective general-purpose standard training procedure.

**Limitations.** DDB's main limitations stem from the high computational cost of training diffusion models. As a trade-off between efficiency and generation quality, we reduce the training image resolution to $64 \times 64$; nonetheless, training our CDPM still requires $\sim 14$ hours on an NVIDIA A30 GPU with 24 GB of VRAM for the largest considered dataset (BFFHQ, 19,200 images). Our ablations show, however, that this cost can be substantially reduced by training for fewer iterations or at lower resolutions (e.g., 3.2 hours for Waterbirds at $32 \times 32$) with only a minor impact on final performance. Importantly, this computational cost should be weighed against the significant, often impractical, burden of manually annotating a bias-aware validation set, which many alternative methods require [34, 43]. Another limitation inherited from diffusion models is that generation quality may decrease if the training dataset is too small, potentially limiting DDB's applicability in very small-scale settings. Finally, while we have benchmarked DDB on a wide range of standard datasets, the bias mitigation community currently lacks million-scale datasets with appropriate bias annotations for large-scale evaluation, which remains an important direction for future research.

**Acknowledgements.** We acknowledge the financial support from PNRR MUR Project PE0000013 "Future Artificial Intelligence Research (FAIR)", funded by the European Union – NextGenerationEU, CUP J33C24000430007, CUP C63C22000770006.

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

# A Technical Appendices and Supplementary Material

## A.1 Datasets

- Waterbirds [43] is an image dataset exhibiting strong correlations ($\rho = 0.950$) between bird species and background environments (*e.g.*, water vs. land). It has 4,795 training images, 1,199 images for validation, and 5,794 for testing.

- BFFHQ [22] builds upon FFHQ [20] by introducing demographic biases for face recognition, while BAR [39] is crafted to challenge action recognition models that associate specific actions with particular stereotypical contexts. For these two last datasets, we specifically refer to their original versions introduced in [39] and [22], respectively: BFFHQ has a total of 19,200 training images where the semantic classes $y \in \{\text{young, old}\}$, the bias attribute $b \in \{\text{female, male}\}$, and the bias ratio $\rho = 0.995$. Then, it provides 1,000 validation images (all bias-aligned) and 1,000 testing images with uniformly distributed labels and bias attributes.

- We utilize the original BAR version as introduced in [39]. The dataset presents 6 different classes, but does not provide explicit bias annotations or a validation set, with 1,941 images for training and 654 for testing.

- ImageNet-A consists of 7,500 real-world images from a 200-class subset of ImageNet, where deep learning models consistently struggle to make correct predictions. Originally introduced by Hendrycks et al. [14] to evaluate adversarial robustness, it is also employed as a model debiasing benchmark, usually as a bias-conflicting evaluation set for models trained on ImageNet-9 [2], a collection of images with 57,200 samples where categories are super-classes of ImageNet, put together to exhibit textural and shape biases [2]. As such, both our CDPM and Bias Amplifier are trained on ImageNet-9. Then, we evaluate both recipes on ImageNet-A in terms of Average Accuracy, following the same protocol of [2, 23, 38].

- UrbanCars has been released in [32] and contains 8000 training images with the target represented by urban and country cars. The dataset is built starting from the original 196 classes of Stanford Cars [26], while synthetically incorporating two different shortcuts, background, from the Places dataset [54], and Co-Occurring Objects (CoObj), from the LVIS one [13]. In UrbanCars, $90.25\%$ of samples present both background and CoObj attributes aligned with the target class biases, $9.5\%$ of training samples are conflicting with respect to only one of the two attributes, while $0.25\%$ are conflicting to both. The evaluation metrics employed for this datasets are the one introduced in [32], namely ID.ACC, BG-Gap, CoObj-Gap, BGCoObj-Gap. ID.ACC is the *in-distribution* accuracy (the weighted per-class accuracy using the training set per-group subpopulation proportions). The other metrics represent the test accuracy gap of each minority subgroup with respect to ID.ACC.

## A.2 Implementation Details

In our experiments, we generate $1000$ synthetic images for each target class using a CDPM implemented as a multi-scale UNet incorporating temporal and conditional embeddings for processing timestep information and class labels. For computational efficiency, input images are resized to $64 \times 64$ pixels. Input images are normalized with respect to ImageNet's mean and standard deviation. The CDPM is trained from scratch for $70,000, 150,000, 200,000, 100.000, 100.000$ iterations for the datasets BAR, Waterbirds, BFFHQ, ImageNet-9, and UrbanCars respectively. Batch size is set to $32$, and the diffusion process is executed over $1,000$ steps with a linear noise schedule (where $\beta_1 = 1e^{-4}$ and $\beta_T = 0.028$). Optimization is performed using MSE loss and AdamW optimizer, with an initial learning rate of $1e^{-4}$, adjusted by a CosineAnnealingLR [36] scheduler with a warm-up period for the first $10\%$ of the total iterations. As per the Bias Amplifier training, we use synthetic images from CDPM. A Densenet-121 [17] with ImageNet pre-training is employed across all datasets, featuring a single-layer linear classification head. The training uses regular Cross-Entropy loss function and the AdamW optimizer [37]. For BFFHQ, BAR, and UrbanCars, the learning rate is $1e^{-4}$, and $5e^{-4}$ for Waterbirds, and ImageNet-9. Training spans 50 epochs with weight decay $\lambda = 0.01$ except for BFFHQ, which uses $\lambda = 1.0$ for 100 epochs. We apply standard basic data augmentation strategies (following [22, 39]), including random crops and horizontal flips. For our debiasing *recipes*, we rely on the same backbones used in the existing literature, *i.e.* ResNet-18 for BFFHQ, BAR, ImageNet-9/A,

and a ResNet-50 for Waterbirds and UrbanCars [39, 22, 25, 32]. In *Recipe II*, $\gamma$ (Equation 3) is set to 3 in all our main experiments. Finally, similarly to [33, 39, 41], we do not rely on a validation set for regularization. This choice is related to real-world application suitability, where no guarantees exist regarding the degree of spurious correlations present in a held-out subset of data used as validation, possibly resulting in detrimental regularization [9].

### A.3 Full Implementation Details of *Recipe I* (two-step)

For the hyperparameters of *Recipe I*, we use for each dataset the following configurations. **BAR**: batch size = 32, learning rate = $5 \times e^{-5}$, weight decay = 0.01, training epochs = 80. **Waterbirds**: batch size = 128, learning rate = $5 \times e^{-4}$, weight decay = 1.0, training for 60 epochs. **BFFHQ**: batch size = 256, learning rate = $5 \times e^{-5}$, weight decay = 1.0, training for 60 epochs. The GDRO optimization is always run with the `-robust` and `-reweight_groups` flags.

For UrbanCars, presenting two biases, we apply Recipe I as it is, considering the filtered correct and wrong classifications in each class as 4 subgroups during GDRO optimization. We then consider all the available 8 subgroups at evaluation time.

### A.4 Full Implementation Details of *Recipe II* (end-to-end)

For **BAR**, we use the same parameters reported in [39]: batch size = 128, learning rate = $1 \times e^{-4}$, weight decay = 0.01, training epochs = 50. **Waterbirds**: batch size = 128, learning rate = $5 \times e^{-5}$, weight decay = 0.01, training epochs = 80. **BFFHQ**: batch size = 256, learning rate = $5 \times e^{-5}$, weight decay = 0.01, training for 50 epochs. For these recipes, we accumulate gradients for 8 iterations (16 for BFFHQ) with mini-batches made of 16 samples.

### A.5 Ablation Studies with DDB's *Recipe II*

This appendix section focuses on additional ablation studies, dedicated to our *Recipe II*. This set of ablations aims at providing additional evidence of Recipe II's robustness and effectiveness, similar to what is shown in Sec. 4.3 of the main paper regarding *Recipe I*. As in the main paper, we run all our ablations on Waterbirds.

Table 5: Ablation on the number of training images for the Bias Amplifier, in terms of WGA on Waterbirds, when using DDB's *Recipe II*.

| # Synth. images for BA training | | | | |
|---|---|---|---|---|
| **# Imgs.** | 100 | 200 | 500 | 1000 | 2000 |
| **WGA** | 84.42 | 90.03 | 90.18 | 90.78 | **91.56** |

**Bias amplifier data requirements.** This ablation study measures how the number of training images for the Bias Amplifier impacts DDB's *Recipe II*. Results are evaluated in terms of WGA on Waterbirds (see Table 4). 200 images (100 images per class) are enough to obtain state-of-the-art competitive results (WGA equal to 90.03), with an increasing number of images corresponding to higher WGA values, up to a maximum of 91.56 for 2000 total images.

Table 6: Impact of guidance strength $w$ on the final WGA on Waterbirds, when using DDB's *Recipe II*.

| Guidance Strength | | | | |
|---|---|---|---|---|
| $w$ | 0 | **1** | 2 | 3 | 5 |
| **WGA** | 87.07 | **91.56** | 88.01 | 89.87 | 88.30 |

**CFG strength and generation bias.** Here, we evaluate DDB's *Recipe II* dependency on the guidance strength parameter. Results are summarized in Table 6. The best results correspond to the usage of $w = 1$, with different guidance strength parameter values having a relatively limited impact on debiasing performance.

**DDB impact on an unbiased dataset.** Similarly to what is shown in Sec. 4.3, we want to ensure that applying *Recipe II* on an unbiased dataset does not degrade the final performance. Also *Recipe II* demonstrates to be applicable in unbiased scenarios, as it measures a final test accuracy of $83.87\%$ on CIFAR10, which is slightly higher than the ERM baseline ($83.26\%$).

**DDB with less biased datasets.** We evaluated DDB's robustness on datasets with weaker spurious correlations by resampling Waterbirds to create versions with $\rho \in \{0.90, 0.80, 0.70\}$. Table 7 shows that while the ERM baseline improves as the bias weakens, DDB-I consistently provides a significant performance boost across all bias levels, demonstrating its effectiveness even when the bias is less pronounced.

Table 7: WGA (%) on Waterbirds with varying bias correlation ($\rho$).

| Method | $\rho = 0.95$ | $\rho = 0.90$ | $\rho = 0.80$ | $\rho = 0.70$ |
|---|---|---|---|---|
| ERM | 62.60 | 63.40 | 64.12 | 68.84 |
| DDB-I | **90.81** | **86.91** | **90.19** | **86.14** |

**Performance on an alternative version of BAR with bias annotations.** We tested out method on an alternative version of the BAR dataset, specifically the one introduced in [29] and having $\rho = 0.95$. Table 8 shows how even for this other version of BAR, our method is effective in mitigating bias dependency. When applying our *Recipe-I* we obtain SOTA performance, confirming the good results already obtained for the benchmark dataset in the main paper.

Table 8: Results on the annotated BAR dataset ($\rho = 0.95$).

| Method | Average Accuracy |
|---|---|
| ERM | 82.40% |
| ETF-Debias [46] | 83.66% |
| DisEnt+BE [28] | 84.96% |
| DDB-I (Ours) | **85.36% ± 0.71** |

**Impact of Generated Image Quality.** We studied the trade-off between the visual quality of generated images, computational cost, and debiasing performance on BFFHQ. We trained the CDPM for fewer iterations, resulting in images with higher (worse) Fréchet Inception Distance (FID) scores. As shown in Table 9, performance degrades gracefully as image quality decreases. Even with a CDPM trained for only 170k iterations (FID of 30.80), the final conflicting accuracy remains competitive with the state-of-the-art and takes only $\sim$9.5 hours to train. This supports our claim that photorealistic quality is secondary to capturing the essential bias cues; even images with artifacts contain the necessary signals for our BA to function effectively.

Table 9: Impact of CDPM training iterations on FID and final conflicting accuracy (%) on BFFHQ with Recipe I.

| # CDPM Iters | FID (lower is better) | Conflicting Acc. |
|---|---|---|
| 200k (original) | 22.60 | 74.67 ± 2.37 |
| 170k | 30.80 | 73.70 ± 0.71 |
| 120k | 36.39 | 71.20 ± 1.13 |
| 70k | 41.82 | 67.40 ± 2.55 |

### A.6 Additional Biased Generations

In this Section, we present 100 additional per class generated images for each dataset utilized in the study, as produced by our CDPM. In the Waterbirds dataset (Figure 4), the model adeptly captures and replicates the correlation between bird species and their background. In the BFFHQ dataset (Figure 8), it effectively reflects demographic biases by mirroring gender-appearance correlations. The results from the BAR dataset (Figure 7) exhibit the model's adeptness in handling multiple bias patterns, preserving typical environmental contexts for different actions. Finally, the ImageNet-9 dataset (Figure 6) stereotypes the various classes and their backgrounds well. Overall, the model

generates high-quality samples that are fidelity-biased across all datasets. Our research confirms that the model exhibits bias-learning behavior, as it not only learns but also amplifies existing biases within diverse datasets. We leverage this characteristic to produce synthetic biased data aimed at enhancing the robustness of debiasing techniques.

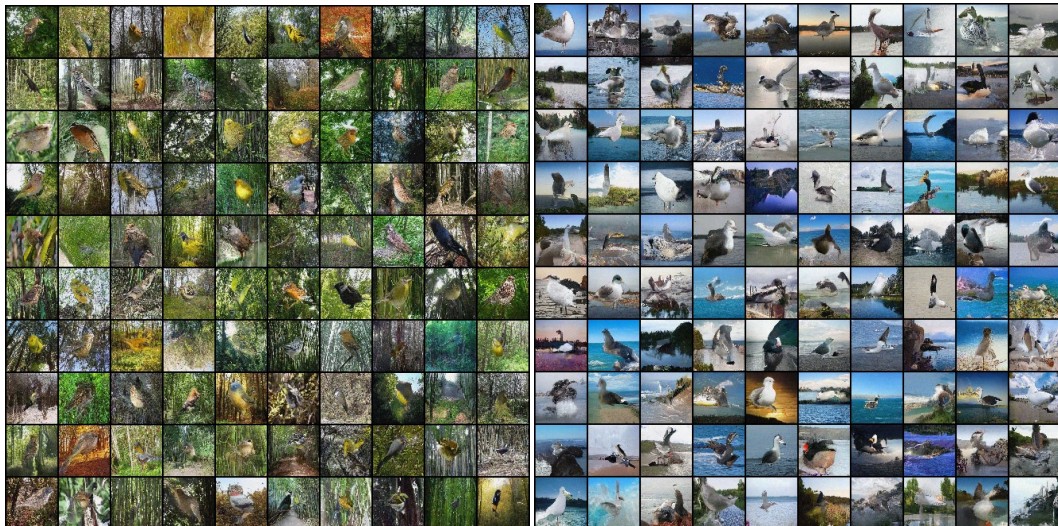

Figure 4: **Synthetic bias-aligned image generations for each class of the Waterbirds dataset.** Each grid displays 100 synthetic images per class, highlighting model bias alignment.

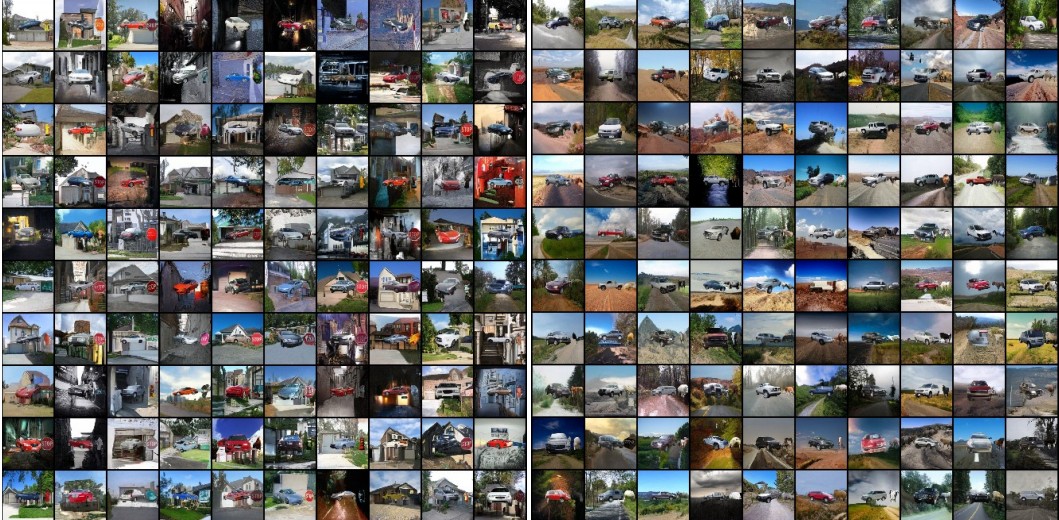

Figure 5: **Synthetic bias-aligned image generations for each class of the UrbanCars dataset.** Each grid displays 100 synthetic images per class, highlighting model bias alignment.

### A.7 Quantitative Analysis of Bias Amplification

To empirically verify our hypothesis that a CDPM amplifies the bias present in a training dataset, we conducted a quantitative analysis on Waterbirds. We created training sets with varying degrees of spurious correlation ($\rho \in \{0.95, 0.90, 0.80, 0.70\}$) and trained a separate CDPM on each. We then generated 1,000 synthetic images per class from each trained model. These synthetic datasets were manually annotated to estimate the resulting bias correlation ($\rho_{synth}$).

As shown in Table 10, the CDPM consistently generated a dataset with a higher bias ratio than the one it was trained on. For example, when trained on data with $\rho = 0.95$, the generated samples

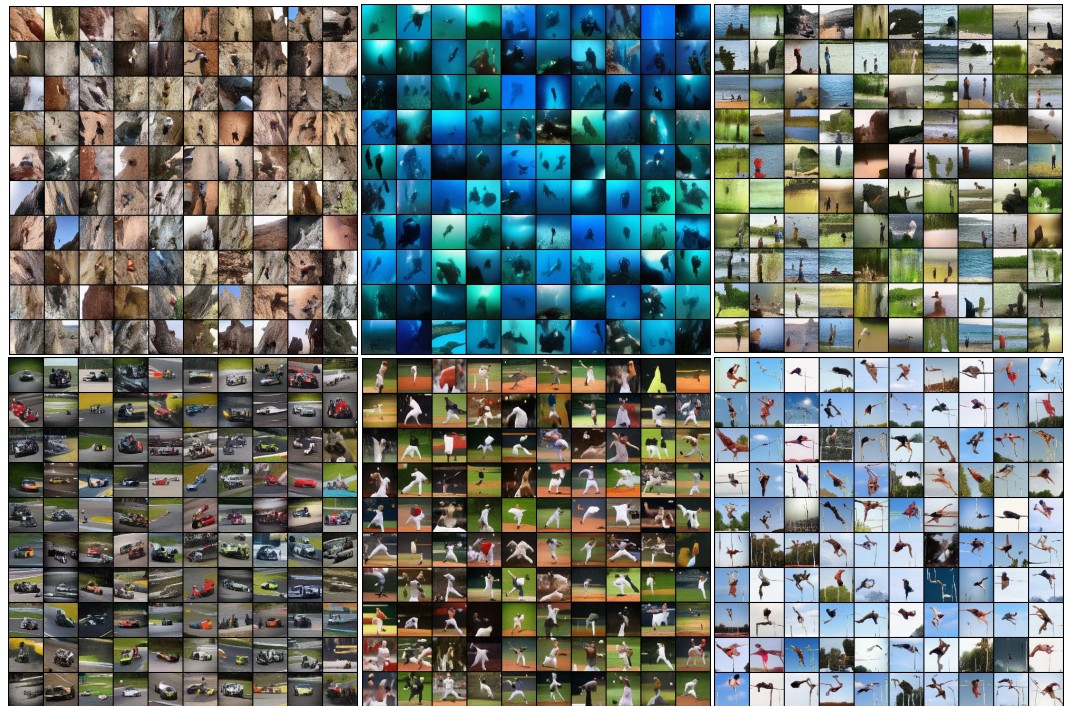

Figure 6: **Synthetic bias-aligned image generations for each class of the BAR dataset.** Each grid displays 100 synthetic images per class, highlighting model bias alignment.

exhibited a correlation of $\rho_{synth} = 0.99$. This empirically demonstrates the bias-amplifying property of diffusion models. This behavior can be attributed to the models learning the density of the training data and, during sampling, having a higher probability of generating samples from the high-density regions [23], which correspond to the majority bias-aligned groups in each class. This amplification is precisely what allows our BA to learn a robust representation of the bias without being exposed to the rare, real bias-conflicting samples.

Table 10: Bias amplification in synthetic data generated by a CDPM trained on Waterbirds with varying training bias ratios ($\rho_{train}$).

| $\rho_{train}$ | 0.950 | 0.900 | 0.800 | 0.700 |
|---|---|---|---|---|
| $\rho_{synth}$ | **0.990** | **0.961** | **0.912** | **0.827** |

## A.8  Bias Validation via Captions in Synthetic Images

As an ulterior confirmation of bias in CDPM-generated images, besides two independent annotators, we implement an unsupervised analysis pipeline using the pre-trained BLIP-2 FlanT5$_{XL}$ model [30] for zero-shot image captioning without any task-specific prompting, for avoiding any guidance in the descriptions of the synthetic data. Our method processes the resulting captions through stop word removal and frequency analysis to reveal underlying biases without relying on predetermined categories or supervised classification models. Figure 9 presents the keyword frequency distribution for the Waterbirds dataset, where the most prevalent terms naturally correspond to known bias attributes (e.g., environmental contexts). These findings confirm that generated images are bias-aligned. Furthermore, the obtained word frequency histograms suggest that images generated by DDB's bias diffusion step may provide information useful to support bias identification (or discovery) at the dataset level.

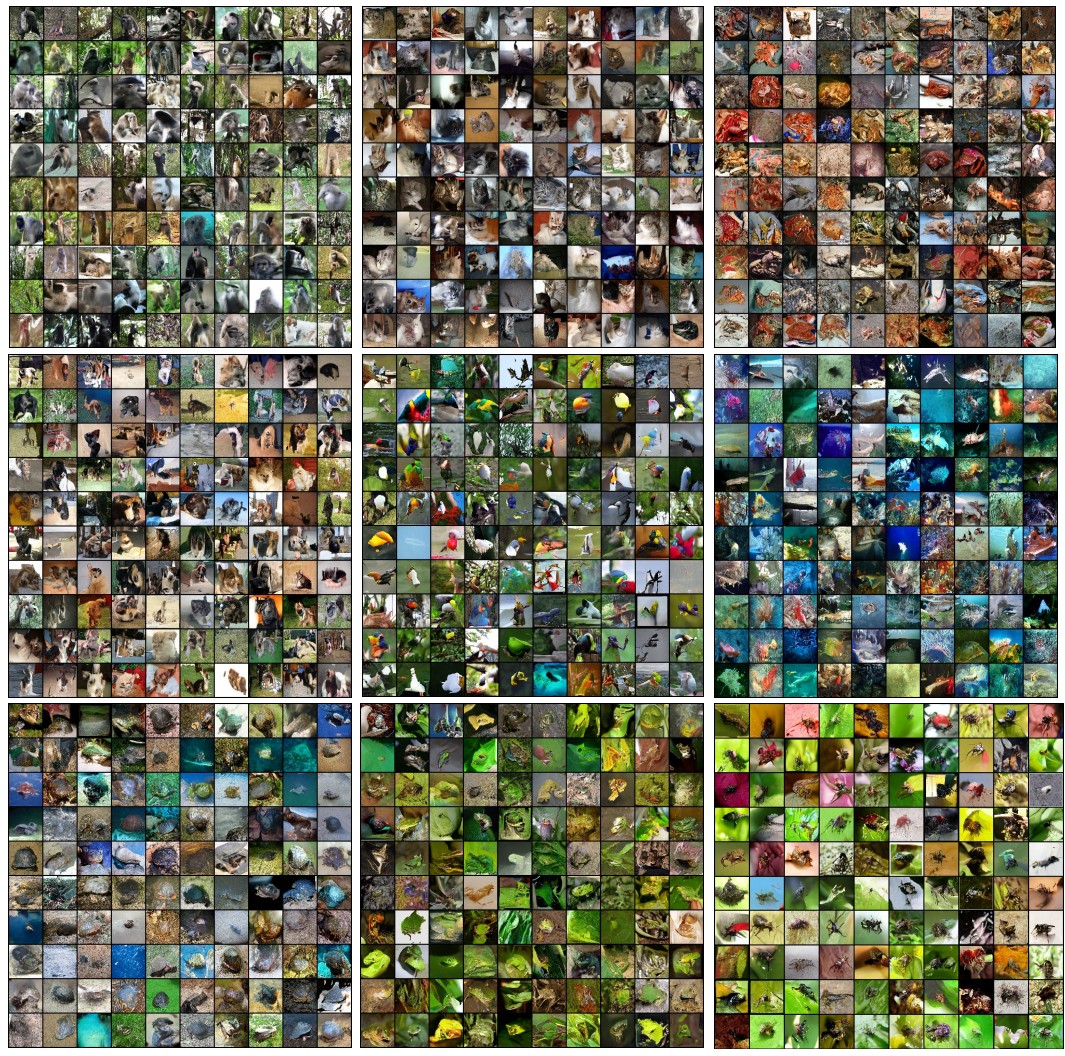

Figure 7: **Synthetic bias-aligned image generations for each class of the ImageNet-9 dataset.**
Each grid displays 100 synthetic images per class, highlighting model bias alignment.

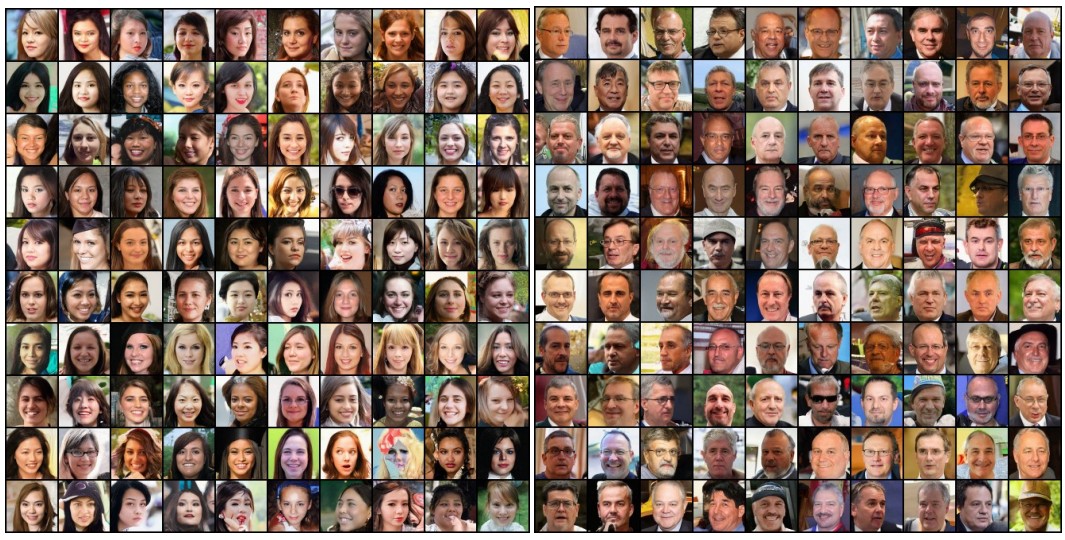

Figure 8: **Synthetic bias-aligned image generations for each class of the BFFHQ dataset.** Each grid displays 100 synthetic images per class, highlighting model bias alignment.

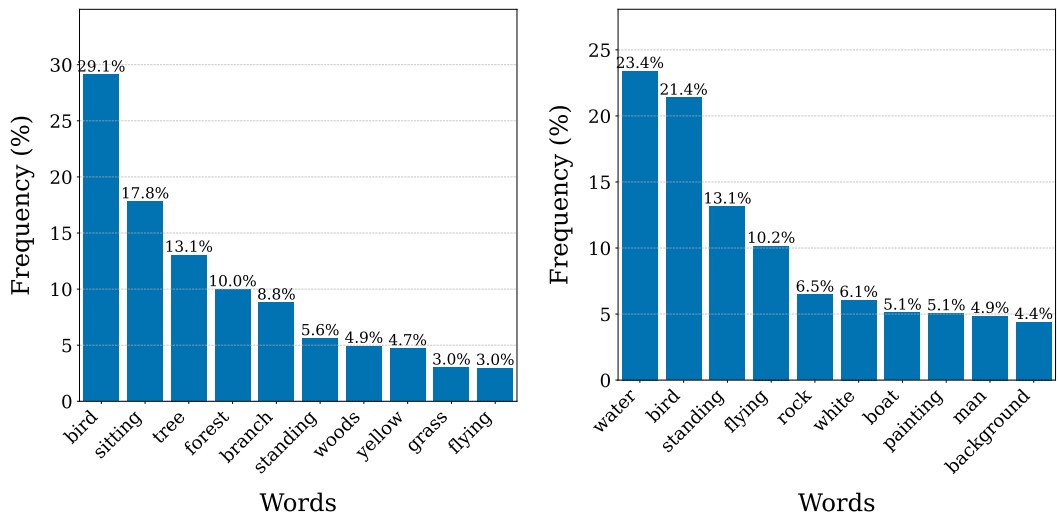

Figure 9: Word frequency analysis from 1000 generated captions for Waterbirds classes 'landbird' (left) and 'waterbird' (right), showing top 10 most frequent terms.

