# OpenReview forum: "Diffusing DeBias: Synthetic Bias Amplification for Model Debiasing"
_NeurIPS.cc/2025/Conference — NeurIPS 2025 poster_

### Official Review · Reviewer_w3kR · 2025-06-23

**Clarity:** 2
**Significance:** 2
**Originality:** 1
**Rating:** 3
**Confidence:** 3

**Summary:**

This paper presents debiasing method using biased data generated by biased diffusion model. Specifically, authors proposed a method to amplify the bias leveraging the bias-aligned samples generated by the diffusion model. The method trains bias amplified auxiliary encoder. Then, the main encoder is debiased by the two different strategies. The result verifies the method.

**Questions:**

Following questions arise:

*1. How does the diffusion model behaves when trained by biased dataset?*

Does the model imitate the bias in dataset, or amplify the bias? Please provide a quantitative analysis for the bias in generated samples.

*2. Quality of generated samples*

Quantitative evaluation for the image quality of generated samples would improve the academic value of this work. If it has degraded visual quality, it is recommended to justify empirical results, at least in conceptual level. Moreover, exploring the properties of the generated samples would enhance the novelty of this work.

**Ethical Concerns:**

["NO or VERY MINOR ethics concerns only"]

**Final Justification:**

I have read authors's responses.

Given that generative model-based approaches are already widely explored in the previous works, the author should provide concrete evidence or analysis to justify the use of a diffusion model over other generative models. In conceptual discussion, authors claim that diffusion model tends to generate samples from majority group, thereby amplifying bias. However, this tendency is also observed in other generation methods, such as GANs, which indicates that the property is not unique to diffusion models.

Overall, the paper employs the diffusion model merely as a tool to generate samples, without introducing any contribution or method to amplify bias of diffusion model. Hence, I carefully determined to maintain my original score.

**Limitations:**

Yes

**Paper Formatting Concerns:**

There is no paper formatting issue.

**Quality:**

2

**Strengths And Weaknesses:**

###  **Strength**
- The method suggests novel method to utilize diffusion model as bias-aligned sample generator.
- The experiments are conducted for various datasets
- Authors provides abundant ablation study to verify the method


-----

### **Weakness**

1. *This work has marginal novelty:*

The proposed work consists of two components, synthesizing the bias-aligned samples by diffusion model, and debiasing main encoder by bias-amplified encoder. However, both main components are based on existing techniques, making the overall method appear as a simple combination of two methods.


2. *More evidence for using diffusion model is recommended:*

Authors utilized the diffusion model to amplify the bias, which is justified only by the citation of previous works. While it is generally understood that diffusion models are sensitive to data bias, an in-depth investigation for the biased diffusion model is recommended to strengthen the originality of this study. Otherwise, it lowers the novelty of this research, as most of contents are from previous works. For instance, section 3.2 in the manuscript introduces diffusion model which is not novel.

It is recommended to further investigate how the diffusion model behaves when trained on biased dataset. Specifically, does the model simply reproduce the existing biases, or does it amplify them beyond what is present in the original data? If the model amplifies the bias, any quantitative analysis demonstrating a bias amplification in the generated samples would be valuable. Such exploration is essential, as understanding the biased sample generated by the diffusion model is central to the motivation of this work and constitutes a core component of the proposed method.

3. *Limited discussion on the generated samples:*

There is limited discussion on the generated bias-aligned samples. They are core in the method, amplifying the bias. Accordingly, in-depth investigations on the synthesized images and their influence to the training are essential. Howerver, the paper only presents few generated samples, which is limited to understand the property of the generated bias-aligned samples. Disussion on the generated samples are limited, for instance, a quantitative analysis on the visual quality. Analysis on the generated samples would help to understand the biased diffusion model, and the proposed framework to reduce the bias.

---

> ### Author Rebuttal · Authors · 2025-07-29
>
> We thank the reviewer for the comments and for recognizing as strengths the novelty of the idea and the quality of the experimental phase including a large ablation analysis. We will provide in the following a punctual response to the raised concerns. References, when applicable, follow the same numbering as in the submitted paper.
>
> **W1. This work has marginal novelty ...**
>
> As also noted by Reviewer 89yi, our main novelty is the usage of diffusion models for amplifying bias, obtaining synthetic samples that can be used for training a bias amplifier instead of using the original training set.
> Unlike existing works employing data augmentations (e.g., BiaSwap [21] and DFA [27]) or adversarial approaches to synthesize bias-conflicting samples (e.g., BiasAdv (Lim et al, CVPR2023)), to the best of our knowledge, this work is the first to propose the usage of diffusion models for actually generating bias-aligned samples, which are to be used for training an auxiliary biased model. Most notable, our approach solves by construction the problem of training set memorization, which affects the proper learning of an effective auxiliary biased model.
>
> Existing methods often employ bias-annotated validation sets or complicated heuristics [34, 43, 51] for deciding when to stop such training. As a bias-annotated validation set is hard to obtain in real-world applications, we argue that solving this problem is a fundamental novelty of this work. This is also supported by the new performed experiments (please, also refer to the first answer to Reviewer 89yi).
>
> Regarding the debiasing recipes, we exploit a popular end-to-end and a two-stage approach to provide evidence on how the proposed method is generally applicable, and how it can potentially be used as an add-on for existing debiasing approaches, providing significant improvements over the original methods.  Furthermore, in this rebuttal, we included additional experiments empirically showing how diffusion models indeed amplify existing biases, and how this actually helps our debiasing recipes.
>
> In the end, we would like to respectfully point out that our work is much more than a simple combination of two known methods. Rather, it has been conceptualized to specifically target known challenges in unsupervised model debiasing, successfully leveraging often overlooked properties of widely used tools such as Diffusion Models in a novel way, and our intuition is also supported by the strong empirical performance.
>
> Regarding Section 3.2, introducing DDPMs and CDPMs (accounting for less than a page, including figures) is intended as a background section necessary to provide the basics that are useful for better understanding the main concepts of the work.
>
>
> **W2 (and Q1). More evidence for using diffusion model is recommended: Authors utilized the diffusion model to amplify the bias ... It is recommended to further investigate how the diffusion model behaves when trained on biased dataset. ... Such exploration is essential ...**
>
> We thank the reviewer for having raised the need of additional analyses to strengthen our contributions. To further highlight that the conditional diffusion model is capturing (and amplifying) the biases of the training set, we manually performed a resampling of the Waterbirds dataset to obtain alternative and less extreme bias ratios ($\rho = 0.90$, $\rho = 0.80$, $\rho=0.70$),  to be used for training our CDPM. From each resulting model, we synthesized 1000 images per class for each $\rho$, with two independent annotators manually labeling bias-aligned/conflicting samples, thus estimating the degree of correlations $\rho$ in the generated samples.  We provide this quantitative analysis in the following table.
>
> | Training $\rho$	| 0.950	| 0.900	| 0.800	| 0.700	|
> |----------------|-----------|-----------|-----------|-----------|
> | **Synthetic** ρ    | 0.990     | 0.961     | 0.912     | 0.827     |
> | **ERM**            | 62.60%    | 63.40%    | 64.12%    | 68.84%    |
> | **DDB-I**          | 90.81%    | 86.91%    | 90.19%    | 86.14%    |
>
>
> As it can be observed, for the original $\rho=0.95$ the CDPM amplifies the bias, and the resulting synthetic training set presents a $\rho$ of 0.99. At the same time, the constructed less biased versions of the datasets also result in the diffusion model amplifying the presence of spurious correlations. We thank the reviewer for suggesting this analysis, and we agree that it adds further value to our work and allows for a better understanding of our main hypotheses and results.
>
>
> **W3 (and Q2). Limited discussion on the generated samples: There is limited discussion on the generated bias-aligned samples. ...**
>
> Please refer to the answer to Reviewer 4iDf, where we provided additional experiments to evaluate the impact of generated image quality over the final debiasing performance.
>
> In summary, in these experiments, we empirically showed that even with an image resolution of 32x32 pixels or with CDPM trained for less iterations (thus increasing the FID of the generated images), we still obtain compelling debiasing performance, as the generated synthetic images still present dominant bias attributes that are effectively captured by our bias amplifier.
>
> For the reduced resolution setting (32x32 pixels instead of 64x64),  when employing our Recipe I (BA + G-DRO) on Waterbirds data, we report a WGA of 90.09% (<1% than the original result of 90.81%), thus measuring a negligible impact on the final performance. In reducing the number of CDPM training iterations from 200k down to 70k, we can observe how, with synthetic images presenting a much higher FID, we are still able to reach SOTA competitive results.

---

> > ### Comment · Reviewer_w3kR · 2025-08-05
> >
> > I appreciate the author's response. However some concerns remain regarding on the diffusion model's behavior on biased datasets. Specifically, how does the bias affects the visual quality of generated samples? Moreover, it is recommended to provide any discussion for the bias-amplifying property of the diffusion models, at least in conceptual level.

---

> > > ### Author Response · Authors · 2025-08-05
> > >
> > > We thank the reviewer for engaging and providing an opportunity to discuss further on this matter.
> > >
> > >
> > > Whether it’s intuitive to assume that cardinality and resolution of original images impact the *visual quality* of generated samples, to our knowledge, no work systematically evaluates the impact of bias on this specific aspect. However, we may expect that bias primarily affects sample diversity rather than visual quality, up to stereotyped generations in the most extreme cases, systematically presenting clear bias attributes, like the examples provided in this work. To complement this response, we computed the FID for the images generated in our experiments involving alternative versions of waterbirds with lower degrees of correlation (0.90, 0.80, and 0.70), noticing only negligible variations in the resulting FIDs (in the order of 1%), suggesting a low impact from bias on image quality.
> > > Nevertheless, further research on this specific topic would be valuable, and we plan to investigate it in future work.
> > >
> > > Regarding bias amplification in diffusion models, on top of the papers already cited in the manuscript [4, 8, 10, 23], we found two additional references regarding how diffusion models can amplify existing imbalances in training data [A, B], which we plan to add and describe in our related work section. For instance, in [A], experiments on datasets of human faces show how, in the generated images, the data imbalances present in the training data regarding both perceived gender and ethnicity are amplified, even if with less extreme effects, as the original distribution was not dramatically skewed like in severe bias scenarios.
> > >
> > > From a high-level perspective, we can consider Diffusion Models as probabilistic models learning the density of the training data. It has been shown that uniformly sampling from these models results in a higher probability of sampling from high-density regions [23, B], which in our case are represented by the predominant bias-aligned populations of each class.
> > > As such, our intuition has been to leverage this behaviour to capture (and amplify) the biased distribution of the training set, so that a bias amplifier could be trained to recognize biased subgroups effectively while being free of memorization issues.
> > >
> > >
> > > We hope that this clarifies the reviewer’s doubts, and we commit to enlarging the discussion regarding this matter in the final version of the manuscript, if accepted.
> > >
> > > [A] Perera, Malsha V., and Vishal M. Patel. "Analyzing bias in diffusion-based face generation models." 2023 IEEE International Joint Conference on Biometrics (IJCB). IEEE, 2023.
> > >
> > > [B] Sehwag, Vikash, et al. "Generating high fidelity data from low-density regions using diffusion models." Proceedings of the IEEE/CVF Conference on Computer Vision and Pattern Recognition. 2022.

---

> > > ### Author Response · Authors · 2025-08-07
> > >
> > > Dear Reviewer w3kR,
> > > Thanks again for your comments and last considerations after our rebuttal.
> > >
> > > In our last reply, we tried to answer your last questions regarding how bias affects the visual quality of generated samples, and we also addressed the bias-amplifying property of the diffusion models.
> > >
> > > We hope that our considerations have been satisfactory for you. If so, we kindly ask you to write a short comment and raise the original rating since, as you well know, the outcome of the evaluation process depends on this and the contribution of each reviewer is relevant. The deadline for revising your review is August 8 AoE, so very close.
> > > If our reply is not yet convincing, we are at your disposal for further discussion.
> > >
> > > We thank you in advance for your work and availability.

---

### Official Review · Reviewer_4iDf · 2025-06-24

**Clarity:** 3
**Significance:** 2
**Originality:** 2
**Rating:** 4
**Confidence:** 4

**Summary:**

The authors proposed Diffusing DeBias, which aims to mitigate the dataset bias by amplifying the bias for the auxiliary biased model with newly generated synthetic data. The generated dataset could be used for methods for both 2-step methods (Group DRO) and end-to-end methods (e.g., LfF). At a high level, bias-aligned samples are generated via classifier-free guidance using a biased training dataset. The authors demonstrated the superiority of the proposed method over existing debiasing methods using 4 different datasets.

**Questions:**

Please refer to the weaknesses.

**Ethical Concerns:**

["NO or VERY MINOR ethics concerns only"]

**Final Justification:**

I read the author's response and my issues are mostly resolved. Although there are two points still remain controversial, I believe leaving them as the limitation and future work would be helpful for the community. Thus, I raised my score from Borderline reject to Borderline accept.

**Limitations:**

Yes

**Quality:**

2

**Strengths And Weaknesses:**

Strengths
- [Scalability] The proposed method could be applied to existing methods that utilize the bias amplification module. Generally, debiasing methods utilize an auxiliary model that focuses on learning the bias attribute. The authors demonstrated that the proposed method could be applied to both 2-step and end-to-end debiasing methods, which seems scalable and convincing.
- [Clarity and organization] The paper is well-written and easy to follow.

Weaknesses
- [Quality of generated images] The visual quality of the generated images, particularly those from real-world datasets such as BAR (Figure 3), appears somewhat unpolished and needs improvement. Including quantitative image quality metrics (e.g., FID, LPIPS) would strengthen the evaluation of the generated images. Furthermore, selectively incorporating high-quality synthetic images based on such metrics could potentially enhance performance. It would be beneficial for the authors to explore whether performance improves by filtering generated samples using the image quality metrics.
- [Computational complexity] The authors stated that high computational complexity is one of the limitations. However, I believe that this limitation is a big issue compared to the performance gain the proposed method brings. I question how practical such an approach would be for removing the bias of large-scale datasets. While the authors tried their best to improve the efficiency by reducing the resolution to 64x64, it degrades the quality of generated images, as stated by the authors. A more thorough discussion or empirical analysis of this trade-off would be valuable.
- [DDB with less biased dataset] Most experiments are conducted on datasets with high bias severity, which facilitates the generation of strongly bias-aligned samples. It remains unclear whether DDB is effective when applied to datasets with subtler or less explicit biases. The authors are encouraged to evaluate the method on datasets with lower bias severity to assess its robustness and generalizability in such settings.
- [Comparisons with image-generation studies] Although the authors reference image-generation-based debiasing methods such as Biaswap [1], they do not include direct comparisons. Given that Biaswap also employs image translation techniques to mitigate bias, a comparative analysis would provide a more comprehensive evaluation of DDB’s relative strengths and limitations. Including such baselines (e.g., ActGen [2]) in the experimental section would significantly improve the paper.

[1] BiaSwap: Removing dataset bias with bias-tailored swapping augmentation (ICCV 2021)

[2] Active Generation for Image Classification (ECCV 2024)

---

> ### Author Rebuttal · Authors · 2025-07-29
>
> We thank the reviewer for the comments, and for appreciating our main idea, its flexibility to be used in both 2-step and end-to-end debiasing methods, as well as the clarity of the exposition. We will now provide a punctual response to the raised concerns. References follow the same numbering as the submitted paper.
>
> **W1 [Quality of generated images] ... and W2 [Computational complexity] ...**
>
> In our experimental stage, we include an extensive and diverse set of benchmark datasets for model debiasing, and we are not aware of a truly large-scale benchmark with proper annotations allowing for bias-mitigation analyses (e.g., equipped with a test set with subgroup annotations). Moreover, we would like to highlight that our performance gain is not negligible, given that we are able to achieve SOTA results across quasi-realistic, real-world data, and multiple bias benchmark datasets, while most of the other (bias) unsupervised debiasing methods struggle in at least one of these settings [31].
>
> As per the resolution of the synthetic generated data, we indeed highlight that we generate images at a resolution of 64x64 pixels to reduce computational time, regardless of the quality of the generated data (as measured by FID or other metrics).
> However, this parameter is not fundamental for our method as we are not interested in perfect fidelity or high quality appearance, but rather in capturing the bias cues in place of the semantic attributes and, as such, we do not mind if some image artifact may be introduced.
> Rather, this is a pro of our approach, as we are not constrained to generate high-quality, perceptually highly realistic images.
>
> To further support this statement, we performed an experiment on Waterbirds data, computing the Fréchet Inception Distance (FID) metric to evaluate the quality of generated images. Specifically, we trained our CDPM on Waterbirds using a pixel resolution of 32x32, obtaining a FID of 58.11, while our original generated images at 64x64 showed a FID of 39.03 (the lower, the better). Exploiting these generated images, we then run our Recipe I (BA + G-DRO): the obtained WGA results 90.09 +/- 0.55% over three independent runs, less than 1% lower than our reported main result of 90.81 +/- 0.68 %.
> The fact that even a low resolution of 32x32 is sufficient to capture the existing shortcuts should further be a reasonable evidence that the quality of the obtained generations is secondary to the final aim of our paper.
>
> As an additional analysis on this matter, we performed new experiments exploiting BFFHQ. Specifically, in our original experiments, we trained the CDPM for 200k iterations, obtaining a FID of 22.60. In these additional experiments, we instead utilize intermediate training iterations, from 70k to 170k. As expected, the images generated with these models show a higher FID, and our aim was to assess how this impacts the final results.
> In the following table, we provide the FID of the generated images and the final resulting Conflicting Accuracy when applying Recipe I (BA+G-DRO).
>
> | # CDPM Iters |   FID   | Conflicting Acc.     |
> |:------------:|:-------:|:---------------------:|
> | 200k (original results)        | 22.60  | 74.67% ± 2.37         |
> | 170k         | 30.80   | 73.70% ± 0.71         |
> | 120k         | 36.39   | 71.20% ± 1.13         |
> | 70k          |  41.82   | 67.40% ± 2.55         |
>
> As we can notice, even if the FID increases by 36% over the original results, when employing 170k training iterations, the final conflicting accuracy is comparable to the one corresponding to images obtained with a model trained for 200k iterations.
>
> Furthermore, it is worth noticing that we achieve competitive results with respect to the state of the art, even when employing 170k iterations, requiring approximately 9.5 hours of training time.
>
> When training the model for 70k iterations only (FID = 41.82), we obtain an average conflicting accuracy of 67.40% (with the best run reaching a 69.2% conflicting accuracy), but requiring less than four hours for training the CDPM. We believe that these results confirm that even if the quality of images decreases, bias is still present in the generated images, and is effectively captured by our BA.
>
>
> **W3. DDB with less biased dataset. Most experiments are conducted on datasets with high bias severity, which facilitates the generation of strongly bias-aligned samples. It remains unclear ...**
>
> We performed dedicated experiments to clarify this matter. Please refer to the answers we provided to reviewers EHbf and w3kR.
>
> In summary, in the first reply, we report experiments on multiple versions of Waterbirds data, which we resampled to have $\rho=0.90$, $\rho=0.80$, $\rho=0.70$. In this experiment, we show how our method is always capable of greatly improving over the ERM baseline. In the second response, we also measure the bias correlation found in the generated images when the CDPM is trained on the same resampled versions of Waterbirds data with less extreme bias, confirming how the presence of bias-aligned samples is always amplified in the resulting synthetic generations.
>
>
> **[W4. Comparisons with image-generation studies. Although the authors reference image-generation-based debiasing methods such as Biaswap ...**
>
> We thank the reviewer for pointing us to other methods for the performance comparison, as well as other works employing generative models to countermeasure generalization failures in image classification (like ActGen).
>
> Regarding the comparison with BiaSwap [21], we provide here a direct comparison between our two debiasing recipes and BiaSwap on two shared benchmark datasets (BAR and BFFHQ).
>
> | Method         | BAR (Average Accuracy) | BFFHQ (Conflicting Accuracy) |
> |:---------------|:----------------------:|:-----------------------------:|
> | BiaSwap        | 52.40%                 | 58.87%                        |
> | DDB-II (ours)  | 72.81%                 | 70.93%                        |
> | DDB-I (ours)   | 70.40%                 | 74.67%                        |
>
> As evidenced by the significant improvement brought by our approach, the intuition of amplifying bias through bias-aligned synthetically generated images is effective and brings more advantages than relying on bias-amplifying loss functions for the auxiliary model (as commonly done in the literature, including BiaSwap, where the GCE [53] is employed), or by attempting to inject synthetic bias-conflicting characteristics into bias-aligned samples (also done in BiaSwap).
>
> For the other suggested reference, ActGen, it does not address the bias mitigation problem and adopts different experimental protocols and data, preventing a direct comparison. Nevertheless, we will incorporate it in the Related Work section of the final version of our manuscript, as improving generalization in challenging downstream tasks through synthetic image generation is a topic linked to ours.

---

> > ### Comment · Reviewer_4iDf · 2025-08-01
> > **Need more discussion on W1 and W2**
> >
> > My concerns regarding W3 and W4 have been adequately addressed. However, I remain unconvinced by the authors’ responses to W1 and W2.
> >
> > The authors argue that the quality of the generated images is of secondary importance, as the primary objective of the work is to enhance debiasing performance. Nonetheless, based on the newly provided table, it appears that higher FID scores, which indicate lower image quality, correlate with degraded performance. While the authors state that the performance remains comparable, the observed trend suggests a performance decline. This apparent contradiction warrants a more thorough explanation. I still think that the quality of generated images plays a crucial role in effective debiasing, and believe that the visual quality of the generated samples presented in the original manuscript still requires improvement.
> >
> > Furthermore, it is difficult to assess whether the reported training time of 9.5 hours is reasonable without knowing the corresponding training time for the baseline methods. I would encourage the authors to include a comparison of training time either per iteration or over the full training schedule across other baselines to provide a clearer context.
> >
> > I appreciate the authors’ efforts and remain open to further discussion. Thank you.

---

> > > ### Author Response · Authors · 2025-08-04
> > >
> > > **W1: On the Quality of Generated Images and its Impact on Performance**
> > >
> > > We apologize for not being clearer on this matter, and we are happy to have the possibility to discuss more.
> > > We agree with the reviewer that a better image quality can be beneficial to the overall final performance. Please note that the purpose of the provided experiment in this rebuttal was to test the sensitivity of our method to lower-quality images, in terms of the final debiasing performance. Indeed, when generating images after only 70k training iterations (which is less than half of the originally employed 200k iterations), the resulting FID almost doubles, and we measure a decline in debiasing performance.
> > >
> > > Our central argument, however, is not that image quality is irrelevant, but rather that perceptual fidelity is secondary to the effective capture of bias signals. Our goal is to generate samples where the bias cue is evident and easily captured by our Bias Amplifier (BA) model. The purpose of these synthetic images is not to be semantically perfect, but to be functionally biased. The fact that performance degrades gracefully with decreasing image quality, rather than catastrophically, supports our claim. Even with an 85% increase in FID (from 22.60 to 41.82 when using the 70k iteration model), the final conflicting accuracy remains competitive and shows a substantial improvement over the ERM baseline (which had an average accuracy of 60.13%). More training (120k and 170k) leads to smaller performance decrease, 3.5% and 1%, respectively, even though FID stays considerably higher (about 36 and 30, respectively).
> > > This empirically demonstrates that even images with noticeable artifacts still contain the salient, low-level cues (e.g., textures, backgrounds, colors) that constitute the dataset's bias, and our BA effectively captures them.
> > >
> > > This supports our intuition of synthesizing bias-aligned samples for training a BA rather than trying to balance the training set by producing more bias-conflicting examples. Whereas synthetic conflicting generations would absolutely require high semantic fidelity to actually help model generalization, in our case the quality we obtain is sufficient for our primary purpose, which is to demonstrate that one can use a conditional generative model to create a substitute and amplified training set for an auxiliary biased model, that can be trained to capture the existing biases without never being exposed to the original training data. In our view, this is a stepping stone to solving the problem of bias-conflicting memorization, as well as determining the optimal stopping point for training an auxiliary biased model—a significant challenge for many existing methods (e.g., [34, 43, 40, 28]), without employing any bias-annotated validation set.
> > >
> > > At the same time, we completely agree with the reviewer that further tuning of the generative model could improve results, but our experiments confirm that the core contribution is effective even without photorealistic generations, yielding state-of-the-art debiasing results with the images shared in our original submission.

---

> > > ### Author Response · Authors · 2025-08-04
> > >
> > > **W2. Contextualizing computational complexity of the proposed method.**
> > >
> > > The reviewer raises a fair point regarding the need to contextualize our training times, and it is true that our initial step has a price to pay in terms of computational burden. However, a direct comparison of model training times can be misleading, as different methods have different prerequisites and associated costs. The ~9.5 hours required to train our CDPM (for BFFHQ) is a fully automated process, which requires no human intervention. In contrast, many state-of-the-art debiasing methods (e.g., JTT [34], CNC[51], GEORGE [43]) rely on a bias-annotated validation set to function optimally. This is used to select the best checkpoint for the biased model or for re-weighting strategies. The process of manually annotating such a set for bias attributes involves significant human labor and time, and its associated cost. Moreover, this may be practically impossible when biases are subtle and not easily categorizable by humans. Our method's computational cost replaces this expensive and often impractical manual annotation step,  presenting a more practical solution, especially in realistic scenarios where clean bias annotations are unavailable.
> > >
> > > Furthermore, there are methods that employ generative models for model debiasing, even though with a different objective (e.g., employing GANs, as in Jung et al., 2023 [18]), sharing a similar computational burden for the first step necessary to train the generative models.
> > >
> > > Additionally, in this rebuttal, we performed an additional experiment to show how further decreasing resolution can improve training time with negligible effect on final accuracy. Specifically, in the rebuttal, we managed to perform an experiment with 32x32 images with the waterbirds dataset, obtaining a comparable WGA (90.09%, with respect to 90.81% of the original experiment using 64x64 image resolution with Recipe I), with the training time reduced to 3.2 hours.
> > >
> > > Finally, it’s worth noting that more advanced strategies for reducing computational burden when using diffusion models exist; however, we did not delve into such approaches, as it was out of scope with respect to our main objective.

---

> > > > ### Comment · Reviewer_4iDf · 2025-08-05
> > > >
> > > > I appreciate the authors' thoughtful response and acknowledge that many of the points raised are valid and well-argued.
> > > >
> > > > That said, I still find the quality of the generated images to be suboptimal, particularly in the examples shown for BAR and ImageNet-9 in Figure 3. Moreover, the computational cost associated with generating these images appears to be non-trivial.
> > > >
> > > > It would strengthen the paper to explicitly acknowledge the limitations related to the image quality and to suggest potential mitigation strategies, such as filtering low-quality samples during training.
> > > >
> > > > In light of the authors' response and the overall contribution, I am willing to revise my score from borderline reject (3) to borderline accept (4).

---

> > > > > ### Author Response · Authors · 2025-08-05
> > > > >
> > > > > We are glad that our responses clarified the reviewer's doubts, and we thank them for the insightful discussion, the useful comments, and for raising the overall score. If accepted, in the final version of the manuscript, we commit to incorporating a discussion on the limitations regarding the quality of the generated images and the suggested additional mitigation strategies.

---

### Official Review · Reviewer_89yi · 2025-06-30

**Clarity:** 2
**Significance:** 3
**Originality:** 3
**Rating:** 4
**Confidence:** 4

**Summary:**

This paper introduces Diffusing DeBias (DDB), a novel framework for unsupervised model debiasing that leverages conditional diffusion models to synthetically generate bias-aligned data. The core idea is to sidestep the contamination introduced by rare bias-conflicting samples in training by training a Bias Amplifier (BA) model solely on synthetically generated, bias-amplified data. DDB is plugged into two debiasing paradigms: a two-step method (DDB-I) and an end-to-end training scheme (DDB-II). The method is evaluated on several standard benchmarks, including Waterbirds, BFFHQ, BAR, and UrbanCars, showing improvements over existing baselines.

**Questions:**

Please see the Weaknesses.

**Ethical Concerns:**

["NO or VERY MINOR ethics concerns only"]

**Final Justification:**

Through the discussion, the authors provided clear responses regarding the consistency between the proposed method and the experimental setup used to support it. They also demonstrated the effectiveness of their approach with well-designed additional ablation studies. Most notably, to the best of my knowledge, the idea of bias amplification through synthesis is novel and has not been previously proposed.

**Limitations:**

The authors properly address the limitations in Section 5.

**Paper Formatting Concerns:**

There is no major formatting issue.

**Quality:**

2

**Strengths And Weaknesses:**

- **Strengths**
    1. The idea of bias amplification through synthesis is a refreshing and original perspective.
    2. The framework’s ability to plug into both two-step and end-to-end approaches enhances its generalizability and potential for adoption.
    3. The experiments are conducted on a diverse set of datasets and baseline methods, supporting the robustness of the evaluation.

- **Weaknesses**
    1. The paper does not appear to include experiments that directly assess the effectiveness of the Bias Amplifier (BA) when trained solely on bias-aligned samples generated by the diffusion model. A comparison between a BA trained on the original training dataset and one trained on synthetic bias-aligned samples—using an otherwise identical debiasing setup—would be valuable. Such an experiment would provide a more rigorous validation of the proposed method’s effectiveness in mitigating bias. (Note: While DDB-II adopts the debiasing strategy of LfF, it differs in that it uses a *frozen* BA to train the debiasing model, whereas LfF jointly trains both components. As such, a direct performance comparison between DDB-II and LfF may not offer clear insights into the impact of synthetic data or the BA training design.)
    2. The performance table presented in Table 1 is not well organized, and nearly half of the cells lack reported performance values.
    3. To the best of my knowledge, the training set of BAR consists solely of bias-aligned samples [1], which—as the authors note—makes it suitable for training a robust bias amplifier. Given this condition, performance differences among debiasing methods on BAR are likely attributable to the use of synthetic data and differing debiasing strategies. Therefore, the BAR dataset may not be well-suited for evaluating the effectiveness of the proposed method (*Diffusing the Bias*).

[1] Nam et al., “Learning from Failure: Training Debiased Classifier from Biased Classifier”, NeurIPS 2020.

---

> ### Author Rebuttal · Authors · 2025-07-29
>
> We thank the reviewer for the insightful comments and for recognizing our main idea as original and refreshing. We will now provide a point-to-point response to the raised concerns. References follow the same numbering as the submitted paper.
>
> **W1. The paper does not appear to include experiments that directly assess the effectiveness of the Bias Amplifier (BA) when trained solely on bias-aligned samples generated by the diffusion model. A comparison between a BA trained on the original training dataset and one trained on synthetic bias-aligned samples—using an otherwise identical debiasing setup—would be valuable. ...**
>
> We thank the reviewer for the comment. Employing Waterbirds (as done with the other ablation studies), we performed the suggested experiments as follows. First, we train the BA on the original dataset, then we employ it for our two Recipes, as follows:
>
> - Frozen BA + LfF (Learning from Failure [38]), that is our Recipe II but with the BA trained on the original real training images (not the diffusion-generated synthetic ones). In this case, we obtain a Worst Group Accuracy (WGA) of 78.45 +/- 2.61 over three runs, with an average decrease of 13.11%. Furthermore, when comparing these results with the original LfF, whose auxiliary biased model is not frozen as the reviewer correctly noted, we got a WGA of 78.00%, which is in line with the result of our new performed experiment, still much less than the performance of our proposed approach.
>
> - With the same BA trained on the original images, we applied our Recipe I, i.e., training GroupDRO with the group annotations obtained as in the Section 3.3.1 of the submitted paper. Here, the results confirm the problem of memorization that we highlighted in our paper. In fact, training the BA on the original images (same setting of our original experiment, which is 50 epochs) results in 100% training accuracy, hence with bias-conflicting samples totally memorized.
> As such, when mining the group pseudo-labels, we end up with two empty sets (corresponding to conflicting in class 0 and class 1). These results confirm that, indeed, memorization is a critical issue when training biased auxiliary models. Nonetheless, to still obtain a somewhat meaningful comparison, we trained the bias amplifier on the original images for only one epoch (as done in JTT [34]), obtaining a final WGA of 79.43 +/- 1.92, again, with a significantly lower mitigation performance than exploiting the BA trained with our original approach.
>
> We will include these results in the final version of our paper, and we thank the reviewer again for suggesting these comparisons to strengthen and add value to our contribution.
>
>
> **W2. The performance table presented in Table 1 is not well organized, and nearly half of the cells lack reported performance values.**
>
> We agree with the reviewer that Table 1 should be improved. The lacking values are due to benchmark methods missing some of the considered datasets. Hence, we plan to divide Table 1 per dataset to have a better organization and readability.
>
>
> **W3. To the best of my knowledge, the training set of BAR consists solely of bias-aligned samples [1], which—as the authors note—makes it suitable for training a robust bias amplifier. ...**
>
> We personally inspected the original version of the BAR dataset (from Nam et al., 2020 [38]), and we can confirm that a few bias-conflicting samples are present in the training set (for instance, see samples with original filenames in the training folder *diving_70, diving_89, diving_99*), with an imbalanced distribution with respect to the classes.
>
> However, as the original version of BAR does not provide bias annotations, we performed additional experiments, employing the BAR version introduced in Lee et al., 2023 [28]. Here, two versions with full bias annotation of BAR are provided (ρ = 0.95 and ρ = 0.99). In the rebuttal time, we managed to perform an experiment on this dataset utilizing the variant with $\rho=0.95$. Here we report the results obtained with our DDB-I (Recipe I), comparing against two SOTA methods on this dataset.
>
>
> Method | Average Accuracy |
> |:---------------------|:---------------:|
> |ERM              | 82.40 |
> |ETF-Debias[46] | 83.66 |
> |DisEnt+BE [28] | 84.96  |
> |DDB-I (Ours) | **85.36 +/- 0.71**|
>
>
> These results confirm the effectiveness of the proposed approach, and we are confident that they can have solved the reviewer's doubts.
>
> We remain at disposal for further information or clarifications on that.

---

> > ### Comment · Reviewer_89yi · 2025-08-06
> >
> > Thank you for your thorough and thoughtful responses. All of my concerns have been clearly addressed. I also reviewed the questions and concerns raised by other reviewers and found the responses to be appropriate and well-considered. Accordingly, I will raise my score.

---

> > > ### Author Response · Authors · 2025-08-07
> > >
> > > We are happy that our rebuttal successfully resolved the reviewer's concerns and that they are willing to raise their score. We also thank them for the thoughtful comments and insights.

---

### Official Review · Reviewer_EHbf · 2025-07-03

**Clarity:** 2
**Significance:** 2
**Originality:** 2
**Rating:** 3
**Confidence:** 4

**Summary:**

This paper introduces Diffusing DeBias (DDB), a novel unsupervised debiasing framework that leverages Conditional Diffusion Probabilistic Models (CDPMs) to synthesize bias-aligned images. These synthetic images are used to train a bias amplifier model, which acts as a supervisory signal in downstream debiasing approaches. The key innovation is to amplify the spurious correlations typically considered harmful in generative models and use this amplified bias as a tool for model debiasing. Two strategies are proposed: 1) DDB-I (Recipe I): A two-step method where the bias amplifier pseudo-labels real training data, which are then used in a Group-DRO optimization. 2) DDB-II (Recipe II): An end-to-end scheme where the bias amplifier guides sample weighting during training. The method is evaluated on several benchmark datasets (Waterbirds, BFFHQ, BAR, UrbanCars, ImageNet-A) and shows strong performance in terms of worst-group accuracy and fairness gaps, often surpassing both supervised and unsupervised baselines.

**Questions:**

Overall, I think the method achieves strong empirical results across a variety of benchmarks. However, concerns remain regarding computational cost, generalization to weaker or more complex biases, and reliance on synthetic data quality. Also, I do believe that it is necessary to perform experiments on larger scale datasets.

**Ethical Concerns:**

["NO or VERY MINOR ethics concerns only"]

**Final Justification:**

I appreciate the authors' rebuttal and the additional results. However, the new evidence is not sufficiently convincing to change my assessment. The method still lacks evaluation on large-scale, non-curated datasets that better reflect real-world distributions, and there is no analysis of robustness across model sizes and training-data. Including these evaluations, or at least acknowledging them as limitations and outlining them as future work, would strengthen the paper.

Moreover, as other reviewers noted, while the authors claim that diffusion models tend to generate samples from majority groups and thereby amplify bias, this tendency is also observed in other generative methods (e.g., GANs), so it is not unique to diffusion models.

Given these unresolved concerns, I will maintain my initial score, and I encourage the authors to further polish the manuscript.

**Limitations:**

Yes

**Quality:**

2

**Strengths And Weaknesses:**

## Strengths

* The Bias Amplifier module can be integrated into both two-step and end-to-end debiasing approaches.
* Empirical Performance: Achieves strong results on a wide variety of benchmarks and bias types (single and multiple spurious correlations).

---

## Weaknesses

* Training diffusion models is expensive (\~14 hours on an A30 GPU for BFFHQ at 64×64 resolution), which may limit scalability.
* The method's effectiveness may diminish when spurious correlations are weak (ρ < 0.95) or when multiple subtle biases coexist.
* *Since synthetic data are sampled from biased distributions, the generated data might lack semantic diversity or introduce synthetic artifacts.
* Limited Real-World Scalability Demonstration: Most experiments are on controlled benchmark datasets. Less is said about generalization to large-scale or long-tail distributions.

---

> ### Author Rebuttal · Authors · 2025-07-29
>
> We thank the reviewer for the comments and for appreciating our empirical performance and the flexibility of the integration of the bias amplifier model. We will now provide a point-to-point response to the raised concerns. References follow the same numbering as the submitted paper.
>
> **Training diffusion models is expensive (~14 hours on an A30 GPU for BFFHQ at 64×64 resolution), which may limit scalability.**
>
> We agree that this is a limitation of our proposed approach, and in fact, we explicitly declared the training time as a possible limitation of our work. However, we also would like to point out that training a diffusion model has nowadays to be considered acceptable and in line with current standards in computer vision and deep learning computational requirements. It is also worth to notice that our reported computational times are referred to a basic implementation of CDPM, as tackling the training time of diffusion models is secondary to our main objective, which aims at showing how a CDPM can be effectively used to generate bias-aligned samples to replace the original training set for training biased auxiliary models, in order to solve by construction some critical open challenges in debiasing scenarios, such as memorization of the few bias-conflicting samples, and when to stop the training of such auxiliary models.
>
> Nonetheless, we reported a new experiment on Waterbirds dataset, where the resolution was reduced to 32x32, obtaining a comparable WGA (90.09%, with respect to 90.81% of the original experiment using 64x64 image resolution with Recipe I), with the training time reduced to 3.2 hours (from an initial training time of roughly 14 hours), showing a significant reduction of the computational time with a negligible decrease of the performance.
> Please refer to the first response to Reviewer 4iDf for more details on this experiment.
>
>
> **The method's effectiveness may diminish when spurious correlations are weak (ρ < 0.95) or when multiple subtle biases coexist.**
>
> In this work, we employ the main benchmark datasets utilized in the image classification model debiasing community. Specifically, all the works on model debiasing that we reference in our paper and benchmark against, exploit a minimum degree of correlation ρ >= 0.95, as the main focus is on mitigating bias in models when the training distribution is predominantly biased. As such, more than a design choice is a convention adopted in this community for benchmarking debiasing algorithms (e.g., see [28, 33, 22, 18, 17, 46, 40]). Furthermore, we can reasonably expect that the generalization of a naively trained ERM model improves naturally when trained on a less severe biased dataset (with ρ < 0.95).
>
> However, we agree that it is interesting to evaluate our DDB approach on a less-biased dataset, and thus, we perform a dedicated set of experiments, resampling the original Waterbirds dataset to obtain three alternative versions with lower amount of training bias-aligned samples (i.e., ρ=0.90, ρ = 0.80, and ρ=0.70). Therefore, we trained both a vanilla model (as a baseline) and our DDB with Recipe I (i.e., bias amplifier + G-DRO), reporting the obtained results in the following table:
> Method | $\rho=0.95$| $\rho=0.90$| $\rho=0.80$| $\rho=0.70$|
> |:-------------|:------------------:|:-----------------------:|:-------------------------:|:------------------:|
> ERM	|	62.60%	|	63.40%		|	64.12%   	|  68.84%	|
> DDB-I 	|90.81%|	86.91%		|	90.19%	|  86.14%|
>
> As we can see, DDB is still capable of obtaining compelling performance, with a significant improvement over the ERM model. Furthermore, the ERM model test accuracy increases when ρ decreases, as expected.
>
> Additionally, in the submitted version of the paper, we provided a dedicated experiment (DDB impact on unbiased dataset, already present in the submitted version) to evaluate the performance of DDB  when no bias is known to be present at all (ρ=0), utilizing CIFAR-10, and obtaining about +1% in accuracy with respect to the ERM.
>
> Finally, regarding multiple biases, we reported the results obtained on UrbanCars, which is a reference benchmark for testing multiple biases in the debiasing community. Specifically, it is constructed to have two different bias attributes (i.e., background and co-occurring objects). The obtained results (following the metrics introduced in the original paper proposing the dataset [31]) show how DDB is actually capable of handling more than one bias. For the latter case, with Recipe I (BA + G-DRO) we manage to reduce the generalization gap with respect to both bias attributes to less than 2\%. In contrast, most of the other unsupervised debiasing methods struggle to mitigate model dependency on both.
>
> We hope that this response clarifies the reviewer’s doubts, and we will include the additional results on Waterbirds in the final version of our manuscript.
>
>
> **Limited Real-World Scalability Demonstration: Most experiments are on controlled benchmark datasets. Less is said about generalization to large-scale or long-tail distributions.**
>
> Connected to our previous answers above, in this work we employed the main benchmark datasets utilized in the image classification model debiasing community (e.g., [28, 33, 22, 18, 17, 46, 40]), which are not large scale. To test for real-world datasets, we utilized BFFHQ and ImageNet-9/A. In this sense, we are not aware of a truly large-scale dataset (e.g., million sized) employed for evaluating debiasing methods.
> About long-tail problems, we argue that evaluating such problems, even if interesting, is not in the scope of this work.
>
>
> **Since synthetic data are sampled from biased distributions, the generated data might lack semantic diversity or introduce synthetic artifacts**
>
> We apologize for our lack of clarity on this point, and we thank the reviewer for raising this comment.
> Indeed, bias is by definition a spurious correlation between target labels and data. In a certain sense, biased (a.k.a. bias-aligned) images may be seen as "stereotyped", with respect to the bias attributes they share.
>
> As such, our initial hypothesis was that, when learning the density of the training distribution with a conditional diffusion model, the bias may dominate the generated images' semantic attributes. Since our final aim is to exploit the generated images to train a bias amplifier, having "stereotyped" images may actually be beneficial for amplifying the bias.
>
> Regarding artifacts, we expect that, when bias attributes are present, a trained model ends up to be strongly relying on bias shortcuts for making its predictions, giving less importance to semantic attributes and potential randomly introduced artifacts.
>
> The reported empirical results support our intuitions, as the usage of DDB makes both our recipes better than the original counterparts, and competitive with existing SOTA methods.

---

> > ### Comment · Reviewer_EHbf · 2025-08-07
> >
> > I appreciate the authors for addressing most of my concerns in the rebuttal.
> >
> > However, I still believe it is important to demonstrate the method’s effectiveness on large-scale, non-curated datasets to better support its real-world applicability. While I understand that such datasets are less commonly used in the debiasing literature, evaluating on more diverse and less controlled distributions would provide stronger evidence for the method’s robustness and scalability.
> >
> > In addition, I encourage the authors to include an analysis of how the method performs across varying model sizes and training data regimes. Demonstrating that the proposed approach is stable and effective across different scales would further strengthen the contribution and its practical relevance.
> >
> > I understand that some of these aspects may be outside the current scope, but acknowledging them as limitations and outlining them as directions for future work would be helpful for the reader.
> >
> > Based on the initial review and the rebuttal from authors, I may keep my initial rating. But i encourage the authors to further polish this paper.

---

> > > ### Author Response · Authors · 2025-08-07
> > >
> > > We are glad that most of the reviewer’s concerns have been addressed by our rebuttal.
> > >
> > > We would like to point out how, in our experiments, we consider several datasets under the following different perspectives:
> > > + Cardinality:  our datasets range from roughly 2000 samples (BAR) and 5000 (Waterbirds) to 20000 (BFFHQ) and 55000 (ImageNet-9/A);
> > > + Imbalance: we consider both perfectly balanced (BFFHQ, UrbanCars) and severely imbalanced datasets (BAR and Waterbirds);
> > > + Semantics: action recognition (BAR), faces (BFFHQ), animals (Waterbirds) and mixed (ImageNet-9/A).
> > > + Multiple Biases (UrbanCars).
> > >
> > > Furthermore, Imagenet-9/A is used to assess the performance of our method on an OOD problem. Overall, we think to have provided a quite large experimental analysis, considering several types of datasets characterized by different properties, confirming the robustness of our proposed approach.
> > >
> > > Finally, we would like to stress again that, in the bias mitigation community, there are no million-size datasets with bias annotations, allowing for a fair state-of-the-art comparison. We will certainly list this as a limitation of current debiasing research, and we will also outline it as an interesting direction for future work.
> > >
> > > We hope that this can solve the reviewer’s concerns, but we are open to further discussion on this matter.

---

### Decision · Program_Chairs · 2025-09-17

**Decision:**

Accept (poster)

**Comment:**

The paper proposes a novel approach to shortcut learning. The core idea is to generate synthetic bias-aligned examples and use them to train a bias-amplifier model, which subsequently guides the training of the downstream classifier.

Reviewers appreciated the conceptual novelty, compelling empirical results, and the clarity of the paper.

There was disagreement about novelty: Reviewer w3kR argued it is marginal given prior generative methods, while others scored novelty higher.

The AC considers the novelty sufficient for NeurIPS. Given the method’s efficacy, focusing on generating biased synthetic examples appears impactful.

Two reviewers remained negative after the rebuttal. Reviewer EHbf emphasized the need for large-scale experiments or multiple-bias datasets. This concern is valid—especially given the field’s shift toward more realistic shortcut settings—but on its own should not preclude acceptance. Concerns about computational cost were partially addressed during the rebuttal.

A thorough comparison to other generative debiasing methods remains a weakness and was only partially addressed during rebuttal.

Reviewer w3kR also emphasized low novelty. Their other two concerns seem less critical for acceptance. First, the lack of a detailed motivation for using a diffusion-based generator is not decisive here. Second, a deeper discussion of generated images would improve the paper, but is not, by itself, a reason to reject.

In summary, the strengths outweigh the weaknesses raised by the reviewers. I recommend accept.